# SYNG4ME: MODEL EVALUATION USING SYNTHETIC TEST DATA

## ABSTRACT

Model evaluation is a crucial step in ensuring reliable machine learning systems. Currently, predictive models are evaluated on held-out test data, quantifying aggregate model performance. Limitations of available test data make it challenging to evaluate model performance on small subgroups or when the environment changes. Synthetic test data provides a unique opportunity to address this challenge; instead of evaluating predictive models on real data, we propose to use synthetic data. This brings two advantages. First, supplementing and increasing the amount of evaluation data can lower the variance of model performance estimates compared to evaluation on the original test data. This is especially true for local performance evaluation in low-density regions, e.g. minority or intersectional groups. Second, generative models can be conditioned as to induce a shift in the synthetic data distribution, allowing us to evaluate how supervised models could perform in different target settings. In this work, we propose SYNG4ME: an automated suite of synthetic data generators for model evaluation. By generating smart synthetic data sets, data practitioners have a new tool for exploring how supervised models may perform on subgroups of the data, and how robust methods are to distributional shifts. We show experimentally that SYNG4ME achieves more accurate performance estimates compared to using the test data alone.

## 1 INTRODUCTION

For machine learning (ML) to be truly useful in safety-critical and high-impact areas such as medicine or finance, it is crucial that models are rigorously audited and evaluated. Failure to perform rigorous testing could result in models at best failing unpredictably and at worst leading to silent failures. There are many examples, such as models that perform unexpectedly on certain subgroups (Oakden-Rayner et al., 2020; Suresh & Guttag, 2019; Cabrera et al., 2019b;a) or models not generalizing across domains due to distributional mismatches (Pianykh et al., 2020; Quinonero-Candela et al., 2008; Koh et al., 2021). Understanding such model limitations is vital to imbue trust in ML systems, as well as guide user understanding as to the conditions in which the model can be safely and reliably used.

Many mature industries involve standardized processes to evaluate performance under a variety of testing and/or operating conditions (Gebru et al., 2021). For instance, automobiles make use of wind tunnels and crash tests to assess specific components, whilst electronic component datasheets outline conditions where reliable operation is guaranteed. Unfortunately, current approaches to characterize performance of supervised ML models do not have the same level of detail and rigor. Instead, the prevailing approach in ML is to evaluate only using average prediction performance on a hold-out test set. Average performance on a test set from the same underlying distribution has two clear disadvantages. *(1) No insight into granular performance*: by treating all samples equally, we may miss performance differences for smaller subgroups. Even if we would decide to evaluate performance on a more granular level, we may not have enough real test data to get an *accurate* evaluation for small subgroups. *(2) Ignores distributional shifts*: the world is constantly evolving, hence the setting of interest may not have the same data distribution as the test set. This typically leads to overestimated real-world performance (Patel et al., 2008; Recht et al., 2019).

The community has attempted to address both of these issues. (1) *Granular performance*: Ribeiro et al. (2020) and Röttger et al. (2021) propose using model behavioral testing methods for model evaluation — which manually craft tests of specific model use-cases, and (2) *Distributional shifts*:

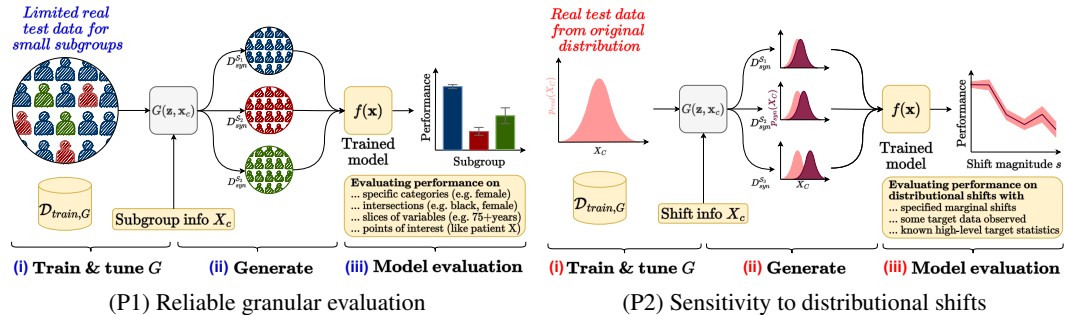

Figure 1: SYNG4ME is a framework for evaluating model performance using synthetic data generators. It has three phases: training the generative model, generating synthetic data and model evaluation. Firstly, SYNG4ME enables (P1) *reliable granular evaluation* when there is (i) limited real test data in small subgroups, by (ii) generating synthetic data conditional on subgroup information $X_c$, thereby (iii) permitting more reliable model evaluation even on small subgroups. Secondly, SYNG4ME enables assessment of (P2) *sensitivity to distributional shifts* when (i) the real test data does not reflect shifts, by (ii) generating synthetic data conditional on marginal shift information of features $X_c$, thereby (iii) quantifying model sensitivity to distributional shift. Required inputs are denoted in yellow.

Koh et al. (2021) and Hendrycks & Dietterich (2018) propose benchmark datasets from different domains for model evaluation—either by finding a real dataset in the wild or through corrupted data. Both approaches are largely manual, labor-intensive processes. This raises the following question:

*Can we define a model evaluation framework where the evaluation is both **low-effort** and **customizable** to new datasets such that practitioners can evaluate their trained ML model(s) under a variety of conditions, customized for their specific tasks and datasets of choice?*

With the above in mind, our goal is to build a model evaluation framework with the following desired properties (P1-P2), motivated by practical, real-world ML failure cases (cited below):

**(P1) Reliable granular evaluation**: we want to accurately evaluate predictive model performance on a granular level, even for regions with few test samples. For example, evaluating performance on (i) *subgroups* (Oakden-Rayner et al., 2020; Suresh et al., 2018; Goel et al., 2020; Cabrera et al., 2019a;b), (ii) *samples of interest* (Deo, 2015; Savage, 2012; Nezhad et al., 2017) and (iii) *low-density regions* (Saria & Subbaswamy, 2019; D'Amour et al., 2020; Cohen et al., 2021).
**(P2) Sensitivity to distributional shifts**: we want to accurately evaluate model performance and sensitivity when the deployment distribution is different, as this often leads to model degradation (Pianykh et al., 2020; Quinonero-Candela et al., 2008; Koh et al., 2021).

**Contributions.** This paper makes the following contributions.

1. **Practical model evaluation framework:** We propose Synthetic Data Generation for Model Evaluation (**SYNG4ME**, pronounced: *"sing for me"*): an evaluation framework for characterizing ML model performance for both (P1) reliable granular evaluation and (P2) sensitivity to distributional shifts (Sec. 4). At its core, SYNG4ME uses generative models to create synthetic test sets for model evaluation. For example, as illustrated in Fig. 1 we can generate larger test sets for small subgroups or test sets with shifts in distribution. To the best of our knowledge, this is the first work that focuses on synthetic data for evaluating supervised models.

2. **Accurate granular performance evaluation:** We find that the use of synthetic test data provides a more accurate estimate of the true performance on small subgroups compared to just using the small (real) test set alone (Sec. 5.1). This is especially true when for evaluating performance on minority and intersectional subgroups, for which we introduce the *intersectional model performance matrix* (Fig. 4).

3. **Quantifying model sensitivity to distributional shifts:** We show how synthetic test data is able to quantify predictive model performance changes as a result of common distributional shifts (defined in Sec. 3), both in terms of model sensitivity across the operating range (Sec. 5.2.1) and with only high-level knowledge of the shift (Sec. 5.2.2).

## 2 RELATED WORK

This paper primarily engages with the literature on model testing and benchmarking, synthetic data, and data-centric AI. We include an extended discussion of related work in Appendix A.

**Model testing.** ML models are mostly evaluated on hold-out datasets, providing a measure of aggregate performance (Flach, 2019). Such aggregate measures do not account for underperformance on specific subgroups (Oakden-Rayner et al., 2020) or assess performance under data shifts (Quinonero-Candela et al., 2008; Wiles et al., 2021). The ML community has tried to remedy these issues. The first approach is to create better benchmark datasets: either synthetic such as Imagenet-C (Hendrycks & Dietterich, 2018) or by collecting real data such as the Wilds benchmark (Koh et al., 2021). Benchmark datasets are labor-intensive to collect and evaluation is limited to specific benchmark tasks, hence this approach is not flexible for *any* dataset or task. The second approach is model behavioral testing of specified properties of an ML model, e.g. see Checklist (Ribeiro et al., 2020) or HateCheck (Röttger et al., 2021). These methods are also labor-intensive, requiring humans to either create or validate the tests. In contrast to both paradigms, SYNG4ME generates the evaluation suite in an automated manner and is applicable to an end-user's specific task.

**Synthetic data.** Improvements in generative models, such as GANs (Goodfellow et al., 2014), have propelled the development of synthetic data. Typically, synthetic data is used to overcome challenges of real data, for example privacy concerns (i.e. to enable data sharing, Jordon et al., 2018; Assefa et al., 2020), and unfairness in real data (Xu et al., 2019a; van Breugel et al., 2021). SYNG4ME provides a different use of synthetic data; improving testing and characterization of ML model performance. This is most closely related to (Antoniou et al., 2017; Dina et al., 2022; Das et al., 2022; Bing et al., 2022), who show that synthetic data can improve downstream model training, especially for making predictions on small subgroups (Bing et al., 2022; Antoniou et al., 2017). However, in contrast to SYNG4ME, these works do not consider distributional shifts (other than balancing the training dataset), and do not explore the use of synthetic data for model *evaluation*.

## 3 PRELIMINARIES

**Notation.** Let $\mathcal{X}$ and $\mathcal{Y}$ be the feature and label space, respectively, together denoted as $\tilde{X} = (X, Y)$. Assume we have a trained black-box prediction model $f : \mathcal{X} \rightarrow \mathcal{Y}$ and also assume we have access to a test dataset $\mathcal{D}_{test,f} = \{x_i, y_i\}_{i=1}^{N_{test,f}}$ with $(x_i, y_i) \overset{iid}{\sim} p(X, Y)$, for underlying distribution $p(X, Y)$. Importantly, we do *not* assume access to the training data of the predictive models, $\mathcal{D}_{train,f}$.

**Generating synthetic datasets for testing.** In the common benchmarking scenario, we compute an average score (e.g. accuracy) of the predictor $f$ over $\mathcal{D}_{test,f}$. However, for our evaluation suite we wish to get an indication of how the model performance may differ for different subgroups or across different parts of the distribution and how it may change if the test distribution is shifted. To evaluate this, we develop a variety of synthetic datasets $\{\mathcal{D}_{syn}^1, \ldots, \mathcal{D}_{syn}^k\}$ based on $\mathcal{D}_{test,f}$, with the desired properties. To do so, we use a deep generative model $G$ trained on $\mathcal{D}_{train,G} = \mathcal{D}_{test,f}$. Note that any existing generative model class can be plugged into this framework, e.g. VAEs, GANs or normalizing flows. Appendix C outlines how the generative model is selected and tuned.

## 4 MODEL EVALUATION USING SYNTHETIC DATA

**Overview.** Our goal is to generate datasets to evaluate predictive models, providing insight into model performance on specific groups or new environments. The latter may consist of data limited to a specific subspace or subgroup, or coming from a shifted distribution. We introduce an evaluation suite *SYNG4ME*, which has the following workflow (Fig. 1): (1) train a (conditional) generative model on the real test set, (2) generate synthetic data conditionally on the subgroup or marginal shift specification, and (3) evaluate model performance on the generated data. As per Sec. 3, the SYNG4ME framework is flexible w.r.t. the generative model. In this paper, we focus on tabular data, the predominant format in many domains, e.g., medicine, finance, manufacturing (Borisov et al., 2021), and hence illustrate SYNG4ME using CTGAN (Xu et al., 2019b) as the generative model—see Appendix C other generative model results and more details on the generative training

process. Below, we elaborate on how SYNG4ME is formulated to enable: (P1) Reliable granular evaluation and (P2) Sensitivity to distributional shifts.

## 4.1 IMPROVING GRANULAR MODEL EVALUATION THROUGH SYNTHETIC DATA (P1)

Models have been shown to have variable performance for different subgroups (Oakden-Rayner et al., 2020; Suresh & Guttag, 2019; Cabrera et al., 2019a;b). Reliable evaluation of model performance at such granular levels is often challenging, especially as we may have access to a limited number of real test examples per subgroup. This challenge is more pronounced with intersectional subgroups.

In evaluating performance on subgroups, we assume that a subgroup $\mathcal{S} \subset \mathcal{X}$ is given. To evaluate the performance of some predictive model $f$ on $\mathcal{S}$, the usual way to assess performance is simply restricting the test set $\mathcal{D}_{test,f}$ to the subspace $\mathcal{S}$. Specifically, given some prediction metric $M : \mathcal{Y} \times \mathcal{Y} \to \mathbb{R}$:

$$A(f; \mathcal{D}_{test,f}, \mathcal{S}) = \frac{1}{\mathcal{D}_{test,f} \cap \mathcal{S}} \sum_{(x,y) \in \mathcal{D}_{test,f} \cap \mathcal{S}} M(f(x), f(y)), \tag{1}$$

which with increasing $|\mathcal{D}_{test,f}|$ converges to the true performance $A^*(f; p, \mathcal{S}) = \mathbb{E}_{X,Y}[M(f(X), Y)|(X, Y) \in \mathcal{S}]$.

**However, what happens when $|\mathcal{D}_{test,f} \cap \mathcal{S}|$ is small?** The variance $\mathbb{V}_{\mathcal{D}_{test,f} \sim p} A(f, \mathcal{D}_{test,f}; \mathcal{S})$ will be large. Therefore, if we use a normal test set for evaluation, this could mean **performance estimates for minority groups are likely to be inaccurate**. Since over- and underestimated performance on these sensitive groups may have negative consequences, this is highly undesirable.

Instead of using $\mathcal{D}_{test,f}$, we use a generative model $G$ trained on $\mathcal{D}_{test,f}$ to create a large synthetic dataset $\mathcal{D}_{syn}$. Subsequently, we measure performance with respect to the synthetic dataset, i.e. $A(f; \mathcal{D}_{syn}, \mathcal{S})$. In Sec. 4.3 we explore why synthetic data may give better estimates compared to test data, i.e. why $|A^* - A(f; \mathcal{D}_{syn}, \mathcal{S})| \le |A^* - A(f; \mathcal{D}_{test,f}, \mathcal{S})|$ in expectation.

**Defining subgroups.** The actual definition of subgroups is flexible. Examples include a specific category of one feature (e.g. female), intersectional subgroups (Crenshaw, 1989) (e.g. black, female), slices from continuous variables (e.g. over 75 years old), particular points of interest (e.g. people similar to patient X), and outlier groups. In Appendix D.1 we elaborate on some of these further.

## 4.2 PREDICTING SENSITIVITY TO DISTRIBUTIONAL SHIFTS (P2)

Distributional shifts between training and test sets are not unusual in practice and have been shown to degrade model performance (Pianykh et al., 2020; Quinonero-Candela et al., 2008; Koh et al., 2021). Unfortunately, often there may be no or insufficient data available from the shifted target domain.

To address this, we can consider a family of shifts $\mathcal{T}$ and test how a model would behave under different shifts in the family. Recall, we consider the shifts defined in Sec. 3. Let $\mathcal{P}$ be the space of distributions defined on $\tilde{\mathcal{X}}$. We test models on data from $T(p)(\tilde{X})$, for all $T \in \mathcal{T}$, with $T : \mathcal{P} \to \mathcal{P}$. The recipe is simple and as follows: (1) Train generator $G$ on $\mathcal{D}_{train,G}$ to fit $p(X)$, (2) Define family of possible shifts $\mathcal{T}$, either with or without background knowledge. Denote shift with magnitude $s$ by $T^s$; (3) Set $s$ and generate data $\mathcal{D}_{syn}^s$ from $T^s(p)$; (4) Evaluate model on $\mathcal{D}_{syn}^s$; (5) Repeat steps 2-4 for different families of shifts and magnitudes $s$.

**Defining shifts.** In practice, we may expect a distributional shift between the training set and target environment (Quinonero-Candela et al., 2008; Amodei et al., 2016; Wu et al., 2021). In some cases, there is prior knowledge to define shifts. For example, covariate shift (Shimodaira, 2000; Moreno-Torres et al., 2012) focuses on a changing covariate distribution $p(X)$, but a constant label distribution $p(Y|X)$ conditional on the features. Label (prior probability) shift (Saerens et al., 2002; Moreno-Torres et al., 2012) is defined vice versa, with fixed $p(X|Y)$ and changing $p(Y)$. [1]

Generalizing this slightly, let $c \subset \{1, ..., |\tilde{X}|\}$ be the indices the features or targets in $\tilde{X}$, of which the marginal distribution may shift. Equivalent to covariate and label shift literature, we assume the distribution $p(\tilde{X}_{\bar{c}}|\tilde{X}_c)$ remains constant ($\bar{c}$ denoting the complement of $c$). [2] Let us denote the

---

[1] Concept drifts are beyond the scope of this work.

[2] This reduces to label and covariate shift for $\tilde{X}_c = Y$ or $\tilde{X}_c = X$, respectively.

marginal's shifted distribution by $p^s(\tilde{X}_c)$ with $s$ the shift magnitude, with $p^0(\tilde{X}_c)$ having generated the original data. The full shifted distribution is $p(\tilde{X}_{\bar{c}}|\tilde{X}_c)p^s(\tilde{X}_c)$.

**Characterizing sensitivity for single marginal shifts.** We consider the typical distributional shift setting, where we assume the marginal of some variables $\tilde{X}_c$ changes, but the distribution of the other variables conditional on $\tilde{X}_c$ remains fixed (see Sec. 3). Without further knowledge, we study the simplest case first: only $\tilde{X}_i$'s marginal is shifted. Letting $p^0(\tilde{X}_i)$ denote the original marginal, let us define a family of shifts $p^s(\tilde{X}_i)$ with $s$ the shift magnitude. For illustration, we choose a mean shift for continuous variables, $p^s(\tilde{X}_i) = p^0(\tilde{X}_i - s)$, and a logistic shift for any binary, logit $p^s(\tilde{X}_i = 1) = \text{logit}\,(p^s(\tilde{X}_i)) - s$. [3] Following Sec. 3, we assume $p(\tilde{X}_{\neg i}|\tilde{X}_i)$ remains constant. This implies $T^s(p)(\tilde{X}) = p^s(\tilde{X}_i)p(\tilde{X}_{\neg i}|X_i)$. This can be repeated for all $i$ and multiple $s$, in order to get characteristic curves of the sensitivity of the model performance to distributional shifts. The actual shift can be achieved using any conditional generative model, with the condition given by $\tilde{X}_i$.

**Incorporating prior knowledge on shift.** In many scenarios, we may want to make stronger assumptions about the types of shift to consider. Let us give two use cases. First, we may acquire high-level statistics of some variables in the target domain—e.g. we may know that the age in the target domain approximately follows a normal distribution $\mathcal{N}(50, 10)$. In other cases, we may actually acquire data in the target domain for some basic variables (e.g. age and gender), but not all variables. In both cases, we can explicitly use this knowledge for sampling the shifted variables $\tilde{X}_c$, and subsequently generating $\tilde{X}_{\bar{c}}|\tilde{X}_c$—e.g. sample (case 1) age from $N(50, 10)$ or (case 2) (age,gender) from the target dataset. Variables $\tilde{X}_{\bar{c}}|\tilde{X}_c$ are generated using the original generator $G$, trained on $\mathcal{D}_{test,f}$.

## 4.3 WHERE THE MAGIC HAPPENS

We should consider why evaluating granular performance on the synthetic data ($A(f; \mathcal{D}_{syn}, \mathcal{S})$) could give better estimates than using just the test data itself ($A(f; \mathcal{D}_{test,f}, \mathcal{S})$)). This is somewhat counterintuitive, since we are training the generative model on the original test set, thereby adding the additional complexity of a generative model and (potentially) no other information (cf. typical data augmentation that benefits from known invariances). The value does *not* come from simply generating more data—a very large synthetic dataset would reduce variance of the downstream predictions w.r.t. the generative model randomness, but not necessarily w.r.t the underlying test distribution from which the generator's data was drawn (which we cannot change).

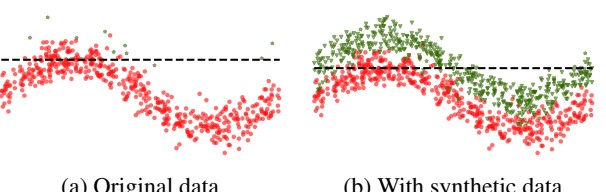

(a) Original data          (b) With synthetic data

Figure 2: **Illustration why synthetic data can give more accurate estimates for low-density regions.** Assume we want to evaluate $f$ (decision boundary=dashed line), which aims to discriminate between $Y = 1$ (green stars) and $Y = 0$ (red circles). Due to the low number of samples for $Y = 1$, evaluating $f$ using the test set alone (Eq. 1) has a high variance. On the other hand, a generative model can learn the manifold from $\mathcal{D}_{test,f}$, and generate additional data for $Y = 1$ by only learning the offset (b, green triangles). This can reduce variance of the estimated performance of $f$.

Instead, the answer may lie in the implicit data representations that the generative model learns (Antoniou et al., 2017); i.e. generative models can learn relationships within the data (e.g. low-dimensional manifolds) from the entire dataset and transfer this knowledge to low-density regions (e.g. small subgroups). We give an example in Fig. 2.

Evidently, synthetic data cannot always help model evaluation; e.g. if there is little structural knowledge that can be transferred from the high- to low-density region, or if there is not enough data to do so. In Appendix C.2, we experimentally explore when synthetic data helps—and when it does not. By combining synthetic data with real data, we observe almost consistent benefits (see Sec. 5).

---

[3]We consider any categorical variable with $m$ classes using $m$ different shifts of the individual probabilities, scaling the other probabilities appropriately.

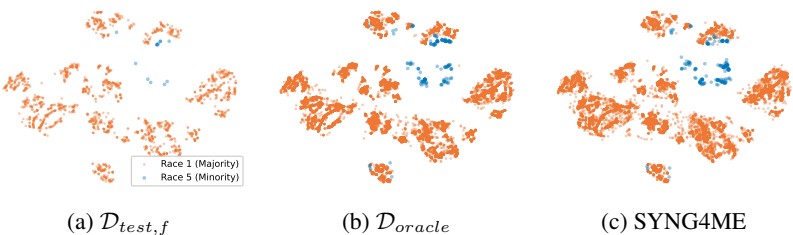

(a) $\mathcal{D}_{test,f}$     (b) $\mathcal{D}_{oracle}$     (c) SYNG4ME

Figure 3: Qualitative assessment of synthetic data. T-SNE on the Adult dataset comparing the real test data $\mathcal{D}_{test,f}$, oracle data $\mathcal{D}_{oracle}$ and SYNG4ME data $\mathcal{D}_{syn}$. We find that SYGN4ME generates synthetic test data that covers the oracle dataset well, despite only having access to $\mathcal{D}_{test,f}$ during training.

## 5  USE CASES OF SYNG4ME

This section demonstrates how SYNG4ME satisfies (**P1**) Reliable granular evaluation and (**P2**) Sensitivity to distributional shifts. We re-iterate that the aim throughout is to estimate the true prediction performance of model $f$ as closely as possible. We tune and select the generative model itself based on Maximum Mean Discrepancy (Gretton et al., 2012), see Appendix C for details.

**Datasets.** We conduct experiments with the following real-world datasets. (**P1**) Reliable granular evaluation (Sec. 5.1) using UCI's *Adult* dataset (Asuncion & Newman, 2007), known to exhibit performance variations between subgroups due to data imbalances (Cabrera et al., 2019a;b) and a *Covid-19 dataset* of Brazilian patients (Baqui et al., 2020), where ethnic subgroups exhibit variations in representation. (**P2**) Sensitivity to distribution shift (Sec. 5.2) using Adult, but also two medical prostate cancer datasets; *SEER* (Duggan et al., 2016) from the USA and *CUTRACT* (Prostate Cancer UK) from the UK, which have the same features, yet have real covariate shift. We describe the datasets, as well as specific experimental details, in Appendix B.

### 5.1  (P1) RELIABLE GRANULAR EVALUATION

**Methodology.** This experiment illustrates the value of synthetic data when evaluating model performance on data subgroups. We consider two types of groups.[4] ① *Minority subgroups*: we evaluate the race subgroup of the Adult dataset, which has severe imbalances in proportional representation of different race groups, with one subgroup accounting for $0.86$ of the data, while the remaining minority subgroups are $< 0.1$ each. We conduct similar ethnicity subgroup evaluation using the Covid-19 dataset (see Appendix D.3.4). The challenge with minority subgroups is that the conventional paradigm of using a hold-out evaluation set might have high-variance in performance estimates due to the small sample size of the subgroup of interest. ② *Intersectional subgroups*: we go beyond a single minority group and evaluate intersectional subgroups (Crenshaw, 1989) (e.g. black males or young females)—for which we introduce the *intersectional model performance matrix* (see Fig. 4).

**Set-up.** We evaluate the subgroup performance of trained model $f$ using different evaluation sets. The baseline is $\mathcal{D}_{test,f}$: a typical hold-out test dataset. We compare this to two SYNG4ME test datasets, which generate to balance the subgroup samples: (i) *SYNG4ME* ($\mathcal{D}_{syn}$): synthetic data generated by $G$, which is trained on $\mathcal{D}_{test,f}$ and (ii) *SYNG4ME+* ($\mathcal{D}_{syn} \cup \mathcal{D}_{test,f}$): test data *augmented* with synthetic dataset. For some subgroup $\mathcal{S}$, each test set gives an estimated model performance $A(f; \mathcal{D}_{\cdot}, \mathcal{S})$, which we compare to the pseudo-oracle performance $A(f; \mathcal{D}_{oracle}, \mathcal{S})$: the performance of model $f$ evaluated on a large amount of unseen real data $\mathcal{D}_{oracle} \sim p(X, Y)$, where $|\mathcal{D}_{oracle}| \gg |\mathcal{D}_{test,f}|$.[5] We desire that the evaluated model performance estimate approximates the model performance on $\mathcal{D}_{oracle}$. This is quantified with the following metrics: (1) *mean absolute error* between $A(f; \mathcal{D}_{oracle}, \mathcal{S})$ and $A(f; \mathcal{D}_{\cdot}, \mathcal{S})$, and (2) *worst-case performance difference* (see Appendix D.3.2).

**Analysis.** Fig. 3 illustrates that the SYNG4ME synthetic data closely resembles the oracle data. Table 1 illustrates that for small subgroups (i.e. racial minorities), SYNG4ME provides a more accurate evaluation of model performance (i.e. with estimates closer to the oracle) compared to a

---

[4]Appendix D.1 contains experiments for other definitions of subgroups, including performance on low-density regions and performance estimates for single samples of interest.

[5]Specifically, $\{\mathcal{D}_{train,f}, \mathcal{D}_{test,f}, \mathcal{D}_{oracle}\} = \{8.4k, 2.1k, 19.6k\}$.

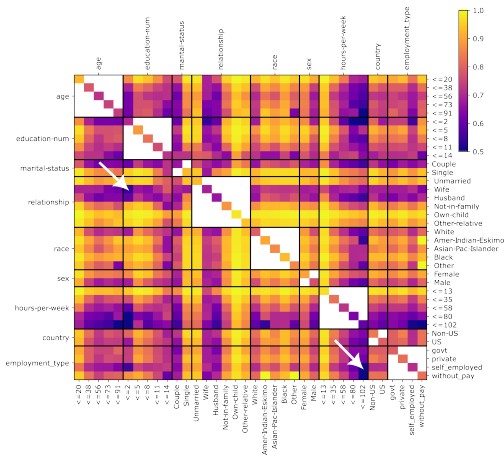

Table 1: Mean absolute difference between predicted performance and performance evaluated by oracle. SYNG4ME better approximates true performance on minority subgroups, compared to test data ($\mathcal{D}_{test,f}$). SYNG4ME+ enjoys the best of both worlds by combining synthetic and real data.

| Model | Subgroup (%) | Mean Absolute Error % ↓ | | |
|---|---|---|---|---|
| | Race | SYNG4ME | SYNG4ME+ | $\mathcal{D}_{test,f}$ |
| RF | #1 (86%) | $7.26 \pm 0.94$ | $2.31 \pm 1.56$ | $10.02 \pm 3.36$ |
| | #2 (9%) | $4.33 \pm 0.34$ | $4.55 \pm 0.38$ | $6.83 \pm 2.67$ |
| | #3 (3%) | $3.48 \pm 0.82$ | $2.98 \pm 0.79$ | $13.68 \pm 4.39$ |
| | #4 (1%) | $1.14 \pm 0.62$ | $1.18 \pm 0.62$ | $7.26 \pm 3.79$ |
| | #5 (1%) | $1.03 \pm 0.85$ | $0.96 \pm 0.84$ | $8.06 \pm 2.00$ |
| GBDT | #1 (86%) | $7.47 \pm 0.83$ | $2.97 \pm 0.85$ | $1.39 \pm 0.98$ |
| | #2 (9%) | $4.25 \pm 0.36$ | $4.07 \pm 0.31$ | $2.14 \pm 1.0$ |
| | #3 (3%) | $4.40 \pm 0.91$ | $4.16 \pm 0.91$ | $4.39 \pm 1.92$ |
| | #4 (1%) | $1.61 \pm 0.62$ | $1.61 \pm 0.65$ | $4.50 \pm 4.73$ |
| | #5 (1%) | $0.68 \pm 0.58$ | $0.68 \pm 0.56$ | $6.31 \pm 3.29$ |
| MLP | #1 (86%) | $6.79 \pm 1.09$ | $2.85 \pm 0.74$ | $1.07 \pm 0.83$ |
| | #2 (9%) | $5.06 \pm 0.37$ | $4.72 \pm 0.27$ | $1.63 \pm 1.35$ |
| | #3 (3%) | $3.60 \pm 0.82$ | $3.43 \pm 0.83$ | $4.75 \pm 0.94$ |
| | #4 (1%) | $0.55 \pm 0.29$ | $0.57 \pm 0.31$ | $6.21 \pm 3.44$ |
| | #5 (1%) | $0.48 \pm 0.53$ | $0.47 \pm 0.54$ | $4.46 \pm 2.85$ |

Figure 4: Intersectional performance matrix for RF model, which diagnoses underperforming (blue) 2-feature subgroups. For example, there is underperformance on "wives" with 2 or less years of education, and self-employed who work more than 80 hours a week (see arrows).

conventional hold-out dataset ($\mathcal{D}_{test,f}$). In addition, SYNG4ME estimates have reduced standard deviation. Thus, despite SYNG4ME using the same (randomly drawn test set) $\mathcal{D}_{test,f}$ to train its generator, its estimates are more robust to this randomness. The results highlight an evaluation pitfall of the standard hold-out test set paradigm: the estimate's high variance w.r.t. the drawn $\mathcal{D}_{test,f}$ could lead to potentially misleading conclusions about model performance in the wild, since an end-user only has access to a single draw of $\mathcal{D}_{test,f}$. e.g., we might incorrectly overestimate the true performance of minorities. The use of synthetic data solves this.

That said, we observe that SYNG4ME sometimes degrades the estimate for the majority group—where there is already enough data for an accurate estimate, the added randomness of the generative model is not outweighed by any potential benefits. SYNG4ME+ achieves the best of both worlds by using a mixture of real and synthetic data, however its majority class performance is still sometimes poorer than that of test data alone. Consequently, we conclude that synthetic data is recommended for small subgroups, but has limited application to subgroups that are already large.

Next, we move beyond single-feature minority subgroups and show that synthetic data can also be used to evaluate performance on **intersectional groups** — subgroups that are typically even smaller due to the intersection. SYNG4ME performance estimates on 2-feature intersections are shown in Fig. 4. Intersectional performance matrices provide model developers insight into where they can improve their model most, as well as inform users how a model may perform on intersections of groups (especially important to evaluate sensitive intersectional subgroups).[6] Appendix E further illustrates how these intersectional performance matrices can be used as part of model reports.

We evaluate the intersectional performance estimates of SYNG4ME and the baseline $\mathcal{D}_{test,f}$ using the Mean Absolute Error of the performance matrices compared to the oracle. The error, averaged across 3 models (i.e, RF, GBDT, MLP), of **SYNG4ME ($0.13 \pm 0.002$)** is significantly lower than $\mathcal{D}_{test,f}$ **($0.21 \pm 0.002$)**, hence demonstrating SYNG4ME provides more reliable intersectional estimates.

**Takeaway.** Synthetic data provides more accurate performance estimates on small subgroups compared to just evaluating on a standard test set. This result coupled with the intersectional model performance matrix is especially relevant from a representational bias and fairness perspective—allowing more accurate evaluation of how models will perform on minority subgroups.

## 5.2 (P2) SENSITIVITY TO DISTRIBUTIONAL SHIFTS

ML models deployed in the wild often encounter data distributed differently from the training set. We simulate distributional shifts in order to evaluate model performance under different operating

---

[6]N.B. low performance estimates by SYNG4ME only indicate poor model performance; this does not necessarily imply that the data itself is biased for these subgroups. However, it could warrant further investigation in potential data bias and how to improve the model.

conditions that might be encountered post-deployment. In the first experiment, we assume no prior knowledge of the shift, whereas in the second we assume some target data is available.

### 5.2.1 NO PRIOR KNOWLEDGE: CHARACTERIZING SENSITIVITY ACROSS OPERATING RANGES

**Methodology.** Assume we have no prior information about the future model deployment environment or how it might change. In this case, we still wish to characterize model behavior and sensitivity for different potential operating conditions, such that a practitioner understands trends of model behavior under different conditions, which can guide as to when the model can and cannot be used.

**Set-up.** We consider shifts in the marginal of some feature $X_i$, keeping $p(\tilde{X}_{\neg i}|X_i)$ fixed (see Sec. 3). We consider a shift in the marginal's mean (see Sec. 4.2). We name the resulting shift-performance plots *model sensitivity curves*; analogous to component characteristic curves widely used in engineering fields. To assess the validity of the behavioral trends captured by model sensitivity curves, we compare estimated performance w.r.t. a pseudo-oracle test set. As in Sec. 5.1, oracle consists of a large unseen test set, which we shift using rejection sampling (see Appendix B).

**Analysis.** To showcase the potential utility, we produce *model sensitivity curves* vs *oracle* (Fig. 5) on two datasets: SEER and Adult. We demonstrate the use of SYNG4ME to understand the potential effect on model performance if the age (Adult) or PSA severity score (SEER) distributions were to shift in mean. The synthetic data curve closely captures the true performance trends across the range of feature shifts. This insight into model performance trends goes much further than "shift = degradation"(Koh et al.,

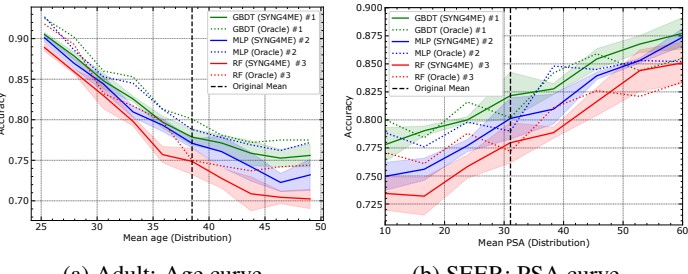

(a) Adult: Age curve      (b) SEER: PSA curve

Figure 5: Model sensitivity curves. SYNG4ME (solid) closely approximates the trends of the oracle (dotted). Model performance ranks to the oracle are retained. Error bars represent variation in performance on data generated over 5 runs. (a) Adult (Age): Performance decreases as mean age increases. (b) SEER (PSA): Performance decreases at lower cancer severity levels (PSA). The x-axis represents marginal's mean.

2021), e.g. some shifts lead to *better* predictions. We also capture the correct model performance ranking, which could help practitioners better understand which model is best for their setting.

**Takeaway.** Synthetic data can be used to generate model sensitivity curves to characterize model performance across the operating range, closely capturing the trends reflected in the oracle data.

### 5.2.2 INCORPORATING PRIOR KNOWLEDGE ON SHIFT

**Methodology.** Consider the scenario where we have *some* knowledge of the shifted distribution. Specifically, here we assume we only observe a few of the features, e.g. age and gender, from the target domain. We sample from this data and generate the other features conditionally (Sec. 4.2).

**Set-up.** We use datasets SEER (US) and CUTRACT (UK), two cancer datasets with the same features, but with shifted distributions. We train models $f$ and $G$ on data from the source domain (SEER). We then wish to estimate likely model performance in the shifted target domain (CUTRACT). We assume access to data from $n$ features in the target domain (features $X_c$), sample $X_c$ from this empirical marginal, conditionally generate $X_{\bar{c}}|X_c$ and evaluate performance on the resulting $\mathcal{D}_{syn}$. To validate our estimate, we use the actual CUTRACT dataset (Target) as ground-truth. As baselines, we use estimates on the source test set, along with *Source Rejection Sampling (RS)*, which achieves a distributional shift through rejection sampling the source data using the observed target features (see Appendix B for details).

**Analysis.** In Fig. 6a, we show the model ranking of the different predictive models based on performance estimates of the different methods. Using the synthetic data from SYNG4ME, we determine the same model ranking as the true ranking on the target—showcasing how SYNG4ME can be used for model selection with distributional shifts. On the other hand, baselines provide incorrect rankings. Fig. 6b shows the average estimated performance of $f$, as a function of the number

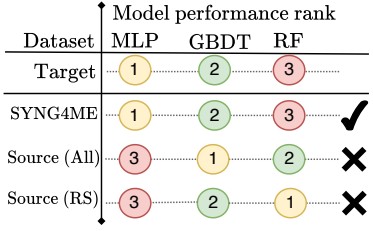

(a) Model performance rank compared.

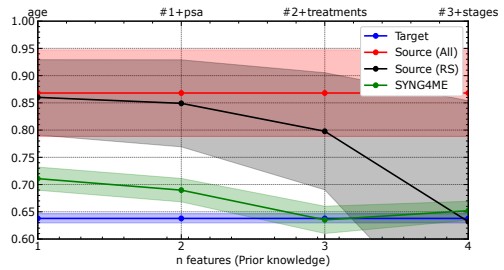

(b) Average accuracy vs increased prior knowledge

Figure 6: Incorporating prior knowledge of the shift. We showcase in (a) $\mathcal{D}_{syn}$ is able to match the performance rank of the true target domain, which can help to select which model is best to use in the target domain and (b) $\mathcal{D}_{syn}$ is better able to approximate performance in the target domain compared to baselines and that performance improves as more prior knowledge is incorporated via added features. *Points are connected to highlight trends.*

of features observed from the target dataset. We see that the SYNG4ME estimates are closer to the oracle across the board compared to baselines. Furthermore, for increasing number of features (i.e. increasing prior knowledge), we observe that SYNG4ME estimates converge to the oracle. This is unsurprising: the more features we observe in the target data, the better we can model the true shifted distribution. Source RS does so too, but more slowly and with major variance issues.

**Takeaway:** Synthetic data can approximate target domain model performance as well as select the best model to use on shifted data when we have high-level information about the potential shift.

## 6 DISCUSSION

**Synthetic data for model evaluation.** Many datasets contain representational bias, in which some sensitive groups (e.g. minority race groups) are poorly represented. In addition, we often wish to understand how models would perform under distributional shifts. We have shown that it is difficult to accurately assess model performance for such subgroups and shifts using available test data, due to a lack of samples. Instead, we have shown that synthetic data can be used to more accurately (and with lower variance) evaluate the performance of a prediction model, even when the generative model is trained on the same test set. This result is surprising and shows the potential value of synthetic data—e.g. SYNG4ME—for model *evaluation* purposes.

**Model reports.** We envision performance estimates using synthetic data could be published alongside models to give insight into when a model should and should not be used. In particular, SYNG4ME could be used to complete model evaluation templates such as Model Cards for Model Reporting (Gebru et al., 2021). Appendix E illustrates an example model report using SYNG4ME.

**Limitations.** Though we have shown that SYNG4ME usually leads to better evaluation compared to simply using a test set, there are limitations to its application. Firstly, SYNG4ME is limited to tabular data. Extending this to other modalities (e.g. text and image) is non-trivial unless annotations are available for meaningfully defining subgroups or shifts, since individual features (e.g. pixels) carry little high-level meaning. Secondly, evaluating the performance under distributional shifts requires assumptions on the shift. These assumptions affect model evaluation and require careful consideration from the end-user. This is especially true for large shifts or other scenarios where we do not have enough training data to describe the shifted distribution well enough. However, even if absolute estimates are inaccurate, we can still provide insight into trend behavior (Fig. 5) and model ranking (Fig. 6). Thirdly, training and tuning a generative model is non-trivial. To counter this, we have included an automatic tuning step in the training process, see Appendix C.

**Including uncertainty in downstream predictions.** Generative models usually do not perfectly approximate the underlying distribution. An interesting avenue for further research is to estimate SYNG4ME's confidence in the performance evaluation, such that an end-user knows when to trust the estimates. In Appendix C.3 we include preliminary results using an ensemble of generative models.

**Training data availability.** In general, we assume no access to any data other than $\mathcal{D}_{test,f}$. However, in some scenarios $\mathcal{D}_{train,f}$, the training data of the predictive models, will be available and could be used to improve the generative model. Even though using this data to train the generative model may induce some bias, this may outweigh the reduced variance. We include results in Appendix D.2.

ETHICS STATEMENT

Like all methods, SYNG4ME relies on the performance of the underlying method - in this case $G$. We have highlighted settings where SYNG4ME should not be used or might not be needed in practice. Assessment of the quality and reliability of the synthetic data is also important, which we cover in Appendix B in terms of both metrics and uncertainty estimates. In this work, we evaluate SYNG4ME using multiple real-world datasets. The Adult dataset is provided by UCI. The three medical datasets namely SEER, CUTRACT and COVID are de-identified and used in accordance with the guidance of the respective data providers.

REPRODUCIBILITY STATEMENT

Further details of the method, experimental setup and datasets are included in Appendices B and C. Code will be released upon acceptance.

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
