# OpenReview forum: "SYNG4ME: Model Evaluation using Synthetic Test Data"
_ICLR.cc/2023/Conference — Submitted to ICLR 2023_

### Official Review · Reviewer_NW99 · 2022-10-24

**Confidence:** 4
**Correctness:** 3
**Technical Novelty And Significance:** 2
**Empirical Novelty And Significance:** 3
**Recommendation:** 6

**Clarity, Quality, Novelty And Reproducibility:**

The paper is very well-written, easy to follow, and the topic it addresses is important and interesting. Using synthetic data to validate machine learning models more carefully and w.r.t. distributional shits seems to be an important direction. I much appreciate that the paper provides a candid discussion toward the limitations and the scope of the proposed approach.

The fact that the method currently only works on tabular data indeed is a strong limitation. Given that there exists severe differences between real and synthetic data in other domains (e.g. for images), it is not clear that the method could easily extend to these areas. The paper also discusses the limitation that generative models for synthetic data may fall short in approximating the real data appropriately. Towards this topic I would appreciate a more in-depth discussion compared to what is currently provided. E.g. I would appreciate pointers to existing work that explores how well generative models are able to approximate tabular data.

Given that the paper introduces a framework rather than an algorithm, enough information is provided to replicate the proposed approach.

**Details Of Ethics Concerns:**

I do not have toward ethical considerations.

**Strength And Weaknesses:**

Strengths:

- The paper addressees an important and interesting problem.
- The reasoning behind the proposed method seems technically sound and the experiments toward granular evaluation and distributional shifts seem reasonable.
- The paper is well-written and easy to follow.
- Limitations of the method are discussed an an appropriate and fair manner.

Weaknesses:

- The paper only focuses on tabular data for which is rather easy to synthetzise data (compare to text and images).

**Summary Of The Paper:**

The proposed paper aims to introduce a framework for validating neural networks with synthetic data. Specifically, the discussed method proposes to train a generative model trained on the commonly used test data to generate a larger and more diverse synthetic dataset. The generated synthetic data is then used to more carefully validate a downstream task model. The argument of the paper is that validating on synthetic data can help lowering the variance of common real validation datasets and to help conditioning generative models to better understand how these models behave w.r.t. different target settings. The paper suggests that validating with synthetic data leads to more accurate performance estimates compared to using real test data alone.

**Summary Of The Review:**

Overall, this is an interesting paper that can be accepted to ICLR 2023. The topic this paper address is important and novel, the paper very well written, and the conducted experiments are sound. Furthermore, the paper carefully discusses the scope and limitations of the proposed method.

---

> ### Author Response · Authors · 2022-11-10
> **Response to reviewer NW99 - Part 4/4**
>
> **References**
>
> [R1] Benjelloun, O., Chen, S. and Noy, N., 2020, November. Google dataset search by the numbers. In International Semantic Web Conference (pp. 667-682). Springer, Cham.
>
> [R2] Sun, B., Yang, L., Zhang, W., Lin, M., Dong, P., Young, C. and Dong, J., 2019. Supertml: Two-dimensional word embedding for the precognition on structured tabular data. In Proceedings of the IEEE/CVF Conference on Computer Vision and Pattern Recognition Workshops (pp. 0-0).
>
> [R3] Kaggle Machine Learning & Data Science Survey, 2017
>
> [R4] Shwartz-Ziv, R. and Armon, A., 2022. Tabular data: Deep learning is not all you need. Information Fusion, 81, pp.84-90.
>
> [R5] Cheng, P., Zhu, H., Tang, X., Liu, D., Chen, Y., Wang, X., Pan, W., Ming, Z. and He, X., 2022. DIWIFT: Discovering Instance-wise Influential Features for Tabular Data. arXiv preprint arXiv:2207.02773.
>
> [R6] Borisov, V., Leemann, T., Seßler, K., Haug, J., Pawelczyk, M. and Kasneci, G., 2021. Deep neural networks and tabular data: A survey. arXiv preprint arXiv:2110.01889.
>
> [R7] Karras, T., Laine, S., & Aila, T. (2019). A style-based generator architecture for generative adversarial networks. In Proceedings of the IEEE/CVF conference on computer vision and pattern recognition (pp. 4401-4410).
>
> [R8] Gulrajani, I., Raffel, C., & Metz, L. (2019). Towards GAN Benchmarks Which Require Generalization. 7th International Conference on Learning Representations, ICLR 2019. https://doi.org/10.48550/arxiv.2001.03653
>
> [R9] Borji, A. (2022). Pros and cons of GAN evaluation measures: New developments. Computer Vision and Image Understanding, 215, 103329. https://doi.org/10.1016/J.CVIU.2021.103329
>
> [R10]  Park, N., Mohammadi, M., Gorde, K., Jajodia, S., Park, H., & Kim, Y. (2018). Data Synthesis based on Generative Adversarial Networks. Proceedings of the VLDB Endowment, 11(10), 1071–1083. https://doi.org/10.14778/3231751.3231757
>
> [R11] Arora, S., Risteski, A., & Zhang, Y. (2018, June 26). Do GANs actually learn the distribution? Some theory and empirics. International Conference on Learning Representations.
>
> [R12] Gretton, A., Borgwardt, K. M., Rasch, M. J., Smola, A., Schölkopf, B., & Smola, A. (2012). A Kernel Two-Sample. Journal of Machine Learning Research, 13, 723–773. www.gatsby.ucl.ac.uk/
>
> [R13] Naeem, M. F., Oh, S. J., Uh, Y., Choi, Y., & Yoo, J. (2020). Reliable Fidelity and Diversity Metrics for Generative Models. In H. D. III & A. Singh (Eds.), Proceedings of the 37th International Conference on Machine Learning (Vol. 119, pp. 7176–7185). PMLR. http://proceedings.mlr.press/v119/naeem20a.html
>
> [R14] Kynkäänniemi, T., Karras, T., Laine, S., Lehtinen, J., & Aila, T. (2019). Improved Precision and Recall Metric for Assessing Generative Models. In Advances in Neural Information Processing Systems (Vol. 32).
>
> [R15] Alaa, A. M., van Breugel, B., Saveliev, E., & van der Schaar, M. (2022). How Faithful is your Synthetic Data? Sample-level Metrics for Evaluating and Auditing Generative Models. Internal Conference on Machine Learning, 290–306. https://arxiv.org/abs/2102.08921v1

---

> ### Author Response · Authors · 2022-11-10
> **Response to reviewer NW99 - Part 3/4**
>
> # (B) Works discussing how well generative models approximate tabular data
>
>     The paper also discusses the limitation that generative models for synthetic data may fall short in approximating the real data appropriately. Towards this topic I would appreciate a more in-depth discussion compared to what is currently provided. E.g. I would appreciate pointers to existing work that explores how well generative models are able to approximate tabular data.
>
> Thank you for raising this point. Measuring the quality of tabular synthetic data is an active area of research (e.g. see [R8, R9]). Numerous generative model works have shown that data generated by generative models preserve utility. For example, in TableGAN [R10] the authors show (Fig. 5) that the train-on-synthetic-test-on-real performance correlates highly with models trained on real data. Another thread of work uses an adversary to test data quality. For example, similar to how GANs are trained, [R11-12] consider how distinguishable the real and synthetic data are by some ML classifier. At last, some authors emphasize that one metric for quality is not enough, and separate quality into diversity and fidelity (e.g. see [R13-15]). In practice, however, it is important to emphasize that the quality of synthetic data depends on the data and generative model.
>
> To this end, in **Appendix C** we explore the influence of (**Appendix C.1**) the generative model choice and tuning, (**Appendix C.3**) uncertainty in the outcome due to possible inaccuracies in the generative modeling process. As we saw in **Section 5.1** and explore further in **Appendix C.2**, in some cases it may not be appropriate to use synthetic data for testing. e.g. when there is already enough real data, such that possible gains are not outweighed by potential inaccuracies in the generative process. Overall, however, the experiments in **Section 5.1** and **Section 5.2**, showcase that for small subgroups and distributional shifts, SYNG4ME provides a useful tool for acquiring more accurate performance evaluations.
>
>
> *We hope this response alleviates your concerns, but please let us know if there are any remaining concerns.*

---

> ### Author Response · Authors · 2022-11-10
> **Response to reviewer NW99 - Part 2/4**
>
> # (A) Applications of tabular data are narrow and ease to generate tabular data.
>
>     SYNG4ME is only capable of handling tabular data. I believe its application scope is too narrow. The fact that the method currently only works on tabular data indeed is a strong limitation.
>
> ------
> **TL;DR:** Tabular data is the most common and ubiquitous data format across many real-world applications compared to other modalities. It is imperative that we build tools to test ML systems with real-world impact reliably. SYNG4ME provides such a solution for tabular data that is applicable to the majority of real-world ML problems.  Tabular data synthesis has unique challenges due to data heterogeneity compared to other modalities, as well as, often small data sizes.
>
> ------
>
> While the focus on tabular data is seemingly limiting, the reality is that **tabular data accounts for the majority of real-world ML applications**—for which reliable testing of models is critical. More specifically, in real-world applications, tabular data is the most common and ubiquitous data format across many fields, such as medicine, finance, manufacturing, e-commerce and climate science, where data is based on relational databases [R2,R4,R5,R6]. This is further highlighted as the Google Dataset platform has around **67% of its 12 million datasets as structured/tabular data** [R1] - which our paper addresses. In addition, Kaggle’s survey of over 16000 data professionals (data scientists, software engineers, data analysts) [R2,R3] found that at least **65% work with tabular data on a daily basis**. In comparison, other modalities are significantly lower, such as images (18%), video (5%), other (11%) etc. If we just consider data scientists, this difference is even more prominent, with 79% working on tabular data daily vs. around 14% on images.
>
> Based on the aforementioned importance and ubiquity of tabular data in real-world ML, it is crucial to ensure that real-world systems that operate with tabular data are reliably tested. SYNG4ME provides a novel contribution for doing so, hence we believe it is relevant for the majority of real-world ML problems.
>
> Furthermore, there are unique challenges associated with tabular data. In contrast to image or language data, tabular data is often heterogeneous with continuous and sparse categorical features [R6]. In addition, tabular datasets are more often modest in size---though image datasets are larger usually consisting of 10,000+ images, tabular data may comprise just a few hundred samples (e.g. UCI’s median dataset size is 1500). This makes the tabular setting challenging compared to modalities such as images. **Finally, a key challenge for tabular data is that although it is easy to find image datasets from different domains, it is hard to find tabular datasets with exactly the same features.**
>
> That being said, we acknowledge the reviewer’s point that it can be easier to generate tabular data given that there are so few training samples because tabular data is often smaller dimensional. That is not to say SYNG4ME cannot be extended to image data, but in this case one may need to resort to using either a pre-trained generative model or an auxiliary dataset with style transfer [R7] to ensure the generated data is realistic for the target distribution. This is beyond the scope of this work.

---

> ### Author Response · Authors · 2022-11-10
> **Response to reviewer NW99 - Part 1/4**
>
> Thank you for your thoughtful comments and suggestions. We give answers to each of the following in turn and highlight the updates to the revised manuscript. In addition, we have uploaded the revised manuscript.  We hope this response alleviates your concerns, but please let us know if there are any remaining concerns.
>
> **(A) Applications of tabular data are narrow and ease to generate tabular data.**
>
> **(B) Works discussing how well generative models approximate tabular data**

---

> ### Author Response · Authors · 2022-11-15
> **Author follow-up**
>
> Dear Reviewer NW99,
>
> We are sincerely grateful for your time and energy reviewing the paper.
>
> We hope that our responses and paper updates have addressed your concerns. Please let us know if you have any outstanding concerns—we would be very happy to address these :)
>
> Thank you!
>
> Paper1873 Authors

---

### Official Review · Reviewer_kuUj · 2022-10-25

**Confidence:** 2
**Correctness:** 3
**Technical Novelty And Significance:** 3
**Empirical Novelty And Significance:** 4
**Recommendation:** 6

**Clarity, Quality, Novelty And Reproducibility:**

The motivation, problem statement and the idea of using synthetic data for test sets looks novel and quite interesting. The positive impact of such evaluation suit on measuring the reliability and robustness of the evaluation metric can be very beneficial to the research community. Overall, paper is written quite neat however some further explanations can transfer the overall idea and purpose of the paper easier.

**Strength And Weaknesses:**

The issues behind the evaluation on small test set is written clearly. The thorough experiments and analysis in the paper and appendix make the idea more convincing.

However, the actual use-cases of such evaluations specifically for small subgroups is not very clear. Is this type of evaluation on synthetic test set is good if we want to get the performance of the metric on small subgroups separately? Since according to table 1 on majority samples the metric tested on D_test,f  is more reliable. Therefore some explanations on why do we want to evaluate on small minorities is beneficial.

In the proposed approach to generate synthetic test set, the generator is trained on a small test set. What is the effect of such a small test set that is used as training dataset on the generator's performance, the newly generated synthetic data and subsequently the performance of the evaluation metric. In other words, from low number of test set how much the generative model is able to learn the manifold from D_test,f.

It is not clear that how much the proposed distributional shiftings in generated synthetic test set are close to the shiftings that could happen during testing.

Similar to section 5.2.1 incorporating some examples in real-word test cases in 5.2.2 also can be very useful.





**Summary Of The Paper:**

This paper proposes an evaluation suite, which generates synthetic test set that helps to better evaluate the performance of supervised learning models for small subgroups (minority groups) and distributional shifts. Augmenting and creating synthetic data enable us to increase the number of data in small subgroups which help to decrease the variance resulting from evaluation metric's performance. Also adding shift to the test distribution make the model more reliable to be tested on different domain shiftings that would happen by testing in different test sets.

**Summary Of The Review:**

This paper and the idea of generating synthetic data and leveraging them for testing classification model can be promising in identifying  reliable metrics. However, there are some weaknesses that mostly are coming from not clear descriptions which should be addressed.

---

> ### Author Response · Authors · 2022-11-10
> **Response to reviewer KuUj - Part 3/3**
>
> # (C) Effect of test set size
>
>     In the proposed approach to generate synthetic test set, the generator is trained on a small test set. What is the effect of such a small test set that is used as training dataset on the generator's performance, the newly generated synthetic data and subsequently the performance of the evaluation metric. In other words, from low number of test set how much the generative model is able to learn the manifold from D_test,f.
>
> The generative model requires enough samples to train and provide accurate synthetic data. What is ``enough’’ will depend on the data characteristics. We explore this, and more, in **Appendix C**. We would like to highlight **Appendix C.3**, in which we provide an approach to estimating the downstream uncertainty **due to the generative process**. This Appendix includes results (Figure 9) on differently-sized $\mathcal{D}_{test,f}$. As expected, when more test data becomes available, the generative model becomes more accurate, in turn giving more accurate estimates of the downstream model performance. More importantly, however, is that this approach can be used in practice to decide whether to trust SYNG4ME’s output. For example, if the test set size is indeed too small, such that the uncertainty in the estimate is large, a practitioner may decide not to trust the SYNG4ME estimate and e.g. instead try to gather more data.
>
> We hope this response alleviates your concerns, but please let us know if there are any remaining concerns.
>
>
> **References**
>
> [R1] Oakden-Rayner, L., Dunnmon, J., Carneiro, G., & Re, C. (2019). Hidden Stratification Causes Clinically Meaningful Failures in Machine Learning for Medical Imaging. ACM CHIL 2020 - Proceedings of the 2020 ACM Conference on Health, Inference, and Learning, 151–159. https://doi.org/10.48550/arxiv.1909.12475
>
> [R2] Suresh, H., & Guttag, J. v. (2019). A Framework for Understanding Sources of Harm throughout the Machine Learning Life Cycle. ACM International Conference Proceeding Series. https://doi.org/10.1145/3465416.3483305
>
> [R3] Cabrera, Á. A., Epperson, W., Hohman, F., Kahng, M., Morgenstern, J., & Chau, D. H. (2019). FairVis: Visual analytics for discovering intersectional bias in machine learning. 2019 IEEE Conference on Visual Analytics Science and Technology (VAST), 46–56.
>
> [R4] Cabrera, Á. A., Epperson, W., Hohman, F., Kahng, M., Morgenstern, J., & Chau, D. H. (2019). FairVis: Visual analytics for discovering intersectional bias in machine learning. 2019 IEEE Conference on Visual Analytics Science and Technology (VAST), 46–56.
>
> [R5] Goel, K., Gu, A., Li, Y., & Ré, C. (2021, August 15). Model Patching: Closing the Subgroup Performance Gap with Data Augmentation. International Conference on Learning Representations. https://doi.org/10.48550/arxiv.2008.06775
>
> [R6] Barocas, S., & Selbst, A. D. (2016). Big data’s disparate impact. Calif. L. Rev., 104, 671.
>
> [R7] Avery, R. B., Brevoort, K. P., & Canner, G. (2012). Does Credit Scoring Produce a Disparate Impact? Real Estate Economics, 40(s1), S65–S114. https://doi.org/https://doi.org/10.1111/j.1540-6229.2012.00348.x
>
> [R8] Bagdasaryan, E., Poursaeed, O., & Shmatikov, V. (2019). Differential Privacy Has Disparate Impact on Model Accuracy. Advances in Neural Information Processing Systems, 32. https://doi.org/10.48550/arxiv.1905.12101
>
> [R9] Adamson, A. S., & Smith, A. (2018). Machine Learning and Health Care Disparities in Dermatology. JAMA Dermatology, 154(11), 1247–1248. https://doi.org/10.1001/JAMADERMATOL.2018.2348

---

> > ### Comment · Reviewer_kuUj · 2022-11-14
> > **Feedback**
> >
> > Thanks for your detailed responses and explanations. By reading them, also the through comparison of your proposed synthetic test data generator versus other two works brought up by the reviewer, the novelty of work and its purpose is more clear. I do encourage to add those differences to the paper.

---

> > > ### Author Response · Authors · 2022-11-15
> > > **Response to feedback + new results**
> > >
> > > ​​Dear Reviewer kuUj,
> > >
> > > We are sincerely grateful for your time and energy reviewing our paper.
> > >
> > > Thank you for your feedback on our response (10 Nov). We have **updated Appendix A** to reflect these comparisons both in text and in **Tables 4 and 5**. We have (15 Nov) also added a similar experiment as in Section 5.2.2, which explores realistic distributional shifts with prior knowledge but with a different dataset (SIVEP on Covid-19 cases in Brazil). Again, we observe that the generated shifted data is realistic when just a few features are observed in the target domain. See **Appendix D.5**, or for a snapshot, see https://imgur.com/a/YFZd5Li
> > >
> > > Please let us know if you have any outstanding concerns---we would be very happy to address these.
> > >
> > > Thank you!
> > >
> > > Paper1873 Authors

---

> ### Author Response · Authors · 2022-11-10
> **Response to reviewer KuUj - Part 2/3**
>
> # (A) Relevance of subgroups evaluation
>
>     [...] the actual use-cases of such evaluations specifically for small subgroups is not very clear. Is this type of evaluation on synthetic test set is good if we want to get the performance of the metric on small subgroups separately? Since according to table 1 on majority samples the metric tested on D_test,f is more reliable. Therefore some explanations on why do we want to evaluate on small minorities is beneficial.
>
> We would like to clarify the importance of evaluation on small subgroups/minorities. Recently, some ML models have gained criticism for poor performance for — or disparate impact on — sensitive groups [R1-R8]. For example, a skin cancer classifier that performs poorly on dark skin, can lead to worrying healthcare disparities across ethnicities [R9].  This highlights, it is of utmost importance that we can reliably estimate model performance (in terms of some metric, like accuracy or direct discrimination) on small subgroups of interest.
>
> As we show, there may not be enough real test data to get an accurate evaluation for small subgroups.
> For example, the intersectional matrix of **Figure 4** cannot be reliably computed using the original test data, because for some indices there may only be a handful of samples. This could lead to incorrect performance estimates. In practice, this implies that the variance of the downstream metric $M$ evaluated on some subset of interest $S$ *w.r.t. the test data draw* is large, due to a small number of samples falling within $S$.
>
> By generating synthetic data, we can reduce this variance. Large subgroups do not benefit from this, because the original test data was already large enough for accurate estimation. As such, we envision SYNG4ME and future synthetic test data generators can provide a tool for gaining more reliable insight into model performance on a granular level.
>
> **Such reliable evaluation is especially relevant from a representational bias and fairness perspective— allowing a more accurate evaluation of how models will perform, even on minority subgroups.**
>
> # (B) Are the generated shifts in synthetic test set close to shifts that could happen during testing?
>
>     It is not clear that how much the proposed distributional shiftings in generated synthetic test set are close to the shiftings that could happen during testing.
>
> Let us start with the setting of **Experiment 5.2.1**, where we have no prior knowledge of the shifts. In this case, all we can do is assume some realistic, simple shifts (e.g. a shift in the mean age), and generate data according to this shift. The main aim of this is to document a model’s behavior, providing insight into how the model performance may potentially change if these shifts were to be observed (i.e. performance over the operating range). The characteristic curves (performance versus shift amplitude) that we introduce, are somewhat similar to the ones found in other fields, e.g. electrical component documentation.
>
> If we do have some knowledge of the shift, we can use this to make the synthetic data more realistic for our specific setting. In **Experiment 5.2.2** we included an experiment with two real-world prostate cancer datasets, SEER (USA) and CUTRACT (UK). Both datasets contain identical features, including information like age, PSA, clinical stage, and received treatment. There is, however, a known distributional shift between these datasets, due to SEER being a US dataset, while CUTRACT contains UK patients. Specifically, UK patients are in general older (mean age of 72 UK vs 69 UK), have lower PSA scores (mean PSA 23 UK vs 31 USA ), have lower prostate cancer stages (mean stage 1.68 UK vs 2.68 USA ) and have fewer comorbidities on average (mean 0.15 UK vs 0.4 USA ).
>
> In this experiment, we only assume to observe some features from the target distribution. This is realistic in practice. For example, for many populations, age and gender statistics are freely available. We model the marginal shift using the observed features in CUTRACT, and generate the other features conditionally using the data from the SEER domain. We see in **Figure 6b** that observing just a few features in the target dataset can help us model the shift. We find that the data that we generate is representative of real CUTRACT, in the sense that we get an accurate estimate of the CUTRACT model performance using just our synthetic test data. At the same time, the closest baseline—using importance sampling to oversample SEER, see Appendix B for details—does not provide a good estimate of the model performance on CUTRACT.

---

> ### Author Response · Authors · 2022-11-10
> **Response to reviewer KuUj - Part 1/3**
>
> Thank you for your thoughtful comments and suggestions. We give answers to each of the following in turn and highlight the updates to the revised manuscript. In addition, we have uploaded the revised manuscript.  We hope this response alleviates your concerns, but please let us know if there are any remaining concerns.
>
> (A) Relevance of subgroup evaluation
>
> (B) Are the generated shifts in the synthetic test set close to shifts that could happen during testing?
>
> (C) Effect of test set size

---

### Official Review · Reviewer_DdMu · 2022-10-25

**Confidence:** 4
**Correctness:** 3
**Technical Novelty And Significance:** 2
**Empirical Novelty And Significance:** 2
**Recommendation:** 5

**Clarity, Quality, Novelty And Reproducibility:**

Clarity: good
Quality: OK
Novelty: low
Reproducibility: low (no code shared)

**Strength And Weaknesses:**

Strengths:

Leveraging synthetic data for AI testing is timely and critical for many tasks, which is the focus of this work.

The paper is clearly written and easy to follow.

The paper considers two scenarios – (1) when the original data suffers from sparse population of important subgroups and (2) when there is a distributional shift between training data and test data.


Weaknesses:
The idea of using synthetic data to evaluate ML models is not entirely new. "Synthetic Data for Social Good" arXiv:1710.08874 has proposed  DataSynthesizer, a privacy-preserving synthetic data generator for creating pathological dataset in tabular domain to be used for model testing.   AITEST described in “Data Synthesis for Testing Black-Box Machine Learning Models” (arXiv:2111.02161) also considers group fairness testing as studied in this paper and leverages goal-oriented synthetic data generation in tabular domain for model testing. AITEST has used data constraint-based data synthesis, in addition to using CTGAN and TVAE as baselines, whereas the current work mainly uses CTGAN. Syng4me considers distribution drift for robustness testing whereas AITEST studies adversarial robustness.


**Summary Of The Paper:**

The paper proposes a framework for using synthetic data to test predictive models for their performance on sparsely populated subgroups and on robustness under distributional shift. The framework focuses on tabular data.

**Summary Of The Review:**

While the paper pursues an important direction, the framework, the methods and the results presented do not provide enough novel insights on using synthetic data for testing fidelity, utility, fairness, and privacy of downstream ML models.

---

> ### Author Response · Authors · 2022-11-10
> **Response to reviewer DdMu - Part 7/7**
>
> **References**
>
> [R1] Xie, L., Lin, K., Wang, S., Wang, F. and Zhou, J., 2018. Differentially private generative adversarial network. arXiv preprint arXiv:1802.06739.
>
> [R2] Song, S., Chaudhuri, K., & Sarwate, A. D. (2013). Stochastic gradient descent with differentially private updates. 2013 IEEE Global Conference on Signal and Information Processing, GlobalSIP 2013 - Proceedings, 245–248. https://doi.org/10.1109/GLOBALSIP.2013.6736861
>
> [R3] Oakden-Rayner, L., Dunnmon, J., Carneiro, G., & Re, C. (2019). Hidden Stratification Causes Clinically Meaningful Failures in Machine Learning for Medical Imaging. ACM CHIL 2020 - Proceedings of the 2020 ACM Conference on Health, Inference, and Learning, 151–159. https://doi.org/10.48550/arxiv.1909.12475
>
> [R4] Suresh, H., & Guttag, J. v. (2019). A Framework for Understanding Sources of Harm throughout the Machine Learning Life Cycle. ACM International Conference Proceeding Series. https://doi.org/10.1145/3465416.3483305
>
> [R5] Cabrera, Á. A., Epperson, W., Hohman, F., Kahng, M., Morgenstern, J., & Chau, D. H. (2019). FairVis: Visual analytics for discovering intersectional bias in machine learning. 2019 IEEE Conference on Visual Analytics Science and Technology (VAST), 46–56.
>
> [R6] Cabrera, Á. A., Epperson, W., Hohman, F., Kahng, M., Morgenstern, J., & Chau, D. H. (2019). FairVis: Visual analytics for discovering intersectional bias in machine learning. 2019 IEEE Conference on Visual Analytics Science and Technology (VAST), 46–56.
>
> [R7] Goel, K., Gu, A., Li, Y., & Ré, C. (2021, August 15). Model Patching: Closing the Subgroup Performance Gap with Data Augmentation. International Conference on Learning Representations. https://doi.org/10.48550/arxiv.2008.06775
>
> [R8] Barocas, S., & Selbst, A. D. (2016). Big data’s disparate impact. Calif. L. Rev., 104, 671.
>
> [R9] Avery, R. B., Brevoort, K. P., & Canner, G. (2012). Does Credit Scoring Produce a Disparate Impact? Real Estate Economics, 40(s1), S65–S114. https://doi.org/https://doi.org/10.1111/j.1540-6229.2012.00348.x
>
> [R10] Bagdasaryan, E., Poursaeed, O., & Shmatikov, V. (2019). Differential Privacy Has Disparate Impact on Model Accuracy. Advances in Neural Information Processing Systems, 32. https://doi.org/10.48550/arxiv.1905.12101
>
> [R11] Timnit Gebru, Jamie Morgenstern, Briana Vecchione, Jennifer Wortman Vaughan, Hanna Wallach, Hal Daume Iii, and Kate Crawford. Datasheets for datasets. ´ Communications of the ACM, 64(12): 86–92, 2021.

---

> ### Author Response · Authors · 2022-11-10
> **Response to reviewer DdMu - Part 6/7**
>
> ``Continuation: (C) Novel insights on using synthetic data for testing``
>
> ## Fairness
> In SYNG4ME, **Section 5.1** we show that generating synthetic test data for small subgroups can yield higher fidelity evaluation of downstream models compared to using the real test set alone. We desire accurate estimates of performance, and over- and underestimated performance estimates on small groups may have negative consequences. Considering small subgroups may correspond to minority groups, we believe this shows SYNG4ME can provide fairer evaluation datasets, in the sense that results are more reliable for everyone. To this end we also presented the intersectional model performance matrix, which provides a simple and accurate way to evaluate how models will perform across intersections of groups. Additionally, SYNG4ME provides test data for distributional shifts, which gives practitioners a new tool for testing how well their models may perform on different populations—which too may be used for anticipating model unfairness. We believe both are (some of the) novel contributions that SYNG4ME offers. We do agree that future work is possible that addresses other issues; for example using fairness metrics as downstream evaluation metric (e.g. how fair will the model be when we shifts the mean age up by 5 years). This can already be achieved using the SYNG4ME method, by replacing the downstream accuracy metric by a fairness metric, including the ones used in AITEST (e.g. group discrimination).
>
> Going further, we show **additional results of how SYNG4ME can be used when computing fairness metrics**. We compare the estimation of fairness metrics using SYNG4ME vs the real test set. Similar to our results in the main paper, we compute the reliability of the estimation of the fairness metrics with respect to the difference to the oracles estimates (i.e. lower is better).
>
> We compute Disparate Impact/Demographic parity and Equalized odds (these metrics are similar to AITEST). The sensitive demographic group we condition on per subgroup is sex. We highlight the results below and include this as an additional experiment in **Appendix D.4**. The results highlight that the usage of synthetic data provides more reliable estimates of the fairness metrics, compared to real test data alone.
>
> ### Disparate impact
>
> Average errors w.r.t oracle (across 5 runs) - lower is better
> | Group    | SYNG4ME      | SYNG4ME+     | Dtest_f      |
> |----------|--------------|--------------|--------------|
> | 1 (86\%) | 0.01 +- 0.01 | 0.01 +- 0.01 | 0.01 +- 0.01 |
> | 2 (9\%)  | 0.03 +- 0.01 | 0.02 +- 0.01 | 0.02 +- 0.02 |
> | 3 (3\%)  | 0.10 +- 0.02 | 0.10 +- 0.02 | 0.12 +- 0.09 |
> | 4 (1\%)  | 0.09 +- 0.01 | 0.09 +- 0.01 | 0.12 +- 0.06 |
> | 5 (1\%)  | 0.03 +- 0.01 | 0.03 +- 0.01 | 0.05 +- 0.04 |
>
> Worst-case error (across 5 runs)  - lower is better
> | Group    | SYNG4ME | SYNG4ME+  | Dtest_f |
> |----------|---------|-----------|---------|
> | 1 (86\%) | 0.033   | 0.02      | 0.021   |
> | 2 (9\%)  | 0.036   | 0.034     | 0.07    |
> | 3 (3\%)  | 0.118   | 0.124     | 0.216   |
> | 4 (1\%)  | 0.102   | 0.102     | 0.193   |
> | 5 (1\%)  | 0.035   | 0.036     | 0.099   |
>
> ### Equalized odds
>
> Average error w.r.t oracle (across 5 runs) - lower is better
> | Group    | SYNG4ME      | SYNG4ME+     | Dtest_f      |
> |----------|--------------|--------------|--------------|
> | 1 (86\%) | 0.18 +- 0.08 | 0.02 +- 0.02 | 0.09 +- 0.07 |
> | 2 (9\%)  | 0.09 +- 0.09 | 0.08 +- 0.04 | 0.53 +- 0.22 |
> | 3 (3\%)  | 0.29 +- 0.08 | 0.31 +- 0.08 | 0.47 +- 0.04 |
> | 4 (1\%)  | 0.12 +- 0.08 | 0.12 +- 0.06 | 0.28 +- 0.24 |
> | 5 (1\%)  | 0.38 +- 0.02 | 0.38 +- 0.02 | 0.46 +- 0.0  |
>
> Worst-case error (across 5 runs)  - lower is better
> | Group    | SYNG4ME | SYNG4ME+  | Dtest_f   |
> |----------|---------|-----------|-----------|
> | 1 (86\%)  | 0.305   | 0.056     | 0.221     |
> | 2 (9\%)   | 0.24    | 0.147     | 0.851     |
> | 3 (3\%)   | 0.377   | 0.393     | 0.549     |
> | 4 (1\%)   | 0.245   | 0.237     | 0.75      |
> | 5 (1\%)   | 0.400     | 0.401     | 0.462     |
>
>
> ## Privacy
> SYNG4ME’s generator could be trained with privacy guarantees, e.g. by replacing the CTGAN by Dp-GAN [R1]  or including differentially private stochastic gradient descent [R2]. This is tangential to our work, hence we have left it for future work.
>
> We hope this response alleviates your concerns. To reflect the discussion above, we have uploaded the revised manuscript. This includes an extended related works section in **Appendix A** and the fairness results in **Appendix D.4**. Please let us know if there are any remaining concerns.

---

> ### Author Response · Authors · 2022-11-10
> **Response to reviewer DdMu - Part 5/7**
>
> # (C) Novel insights on using synthetic data for testing
>
>     While the paper pursues an important direction, the framework, the methods and the results presented do not provide enough novel insights on using synthetic data for testing fidelity, utility, fairness, and privacy of downstream ML models.
>
> ------
>
> **TL;DR:** We address the novel insights on using synthetic data or testing fidelity, utility, fairness, and privacy of downstream ML models. In addition, we present additional results addressing the fairness perspective.
>
> ------
>
> We address the issue of providing novel insights on using synthetic data for testing fidelity, utility, fairness, and privacy of downstream ML models, addressing each in turn.
>
> We provide **additional results** showcasing insights from a fairness perspective below (point 3), and include these in a new **Appendix D.4**
>
>
> ## Fidelity
> In **Appendix C.1** we have outlined how we can assess the fidelity of the synthetic data itself, along with uncertainty quantification in **Appendix C.3** to guide when we can trust performance estimates on the basis of the synthetic test data. In terms of fidelity of the prediction performance, we compare the estimates from SYNG4ME to an oracle dataset. We show that the estimated performance using SYNG4ME is similar to a large unseen real oracle dataset, compared to real test data alone. See experiments in **Sections 5.1 and 5.2**.
>
> ## Utility
> Utility of synthetic data is often assessed by whether it can provide similar results as the original data, and hence this follows to some extent from fidelity. We showcase utility both on the granular subgroup evaluation and distributional shifts. Specifically, in SYNG4ME **Section 5.1** we show that generating synthetic test data for small subgroups can yield higher fidelity evaluation of downstream models compared to using the real test set alone, and in **Section 5.2.1** we show that SYNG4ME matches the same performance trend as when we evaluate using the (shifted) oracle dataset. In **Section 5.2.2** we add a second real dataset and show that by incorporating some information of this data (specifically observing some of the features), we can generate synthetic data that provides the same performance estimates as the real (largely unobserved) target dataset. Hence, these results highlight the utility of synthetic data in providing accurate estimates of true performance.
>
> ``Continues in next part``

---

> ### Author Response · Authors · 2022-11-10
> **Response to reviewer DdMu - Part 4/7**
>
> ``Continuation: (B) Comparison to other tabular data approaches``
>
> ## AITEST
>
> We will include AITEST in the related work in the **updated Appendix A**, since indeed it uses synthetic data for testing. Let us briefly outline why we believe SYNG4ME is novel and significantly different compared to AITEST. We contrast SYNG4ME to AITEST in terms of aims and assumptions, algorithm, and use cases.
>
> ### Aims and assumptions
> AITEST has a significantly different aim and method compared to SYNG4ME. As mentioned by the reviewer, AITEST can test for adversarial robustness by generating realistic data with user-defined constraints, but this is different from our work that aims to generate synthetic test data for granular evaluation and distributional shifts.
>
> Additionally, the assumptions on user input are quite different: AITEST enables users to define constraints on features and associations between features, whereas SYNG4ME requires information in terms of which subgroups to test or shifts to generate.  We do see possibilities to combine both frameworks, e.g. through including constraints similar to the ones AITEST uses within the SYNG4ME method, or using fairness as a downstream task.
>
> We have taken a step in this direction and added fairness as an additional experiment (**See the next point for the fairness experiment**) and have included this experiment in the **new Appendix D.4**. We will also include the discussion of AITEST in the related work in **updated Appendix A**.
>
> ### Algorithmic
> AITEST requires a decision tree surrogate of the black-box model, whilst SYNG4ME does not need to model the black-box predictive model.
> AITEST defines data constraints by fitting different distributions to the features and using statistical testing to select the correct distribution. The dependencies are then captured by a DAG. SYNG4ME does not require predefined constraints and dependencies, but aims to learn these implicitly with the generative model.
>
>
> ### Use cases
>
> - Group fairness:  AITEST aims to probe if a model does have a group fairness issue or not. The goal of SYNG4ME is different — even if models don’t have group bias issues, with SYNG4ME we desire reliable performance metric estimates (accuracy or even fairness) which are similar to the oracle estimates on small and intersectional subgroups for which we have limited real test data.
> - Adversarial robustness: AITEST does this by generating more inputs in the neighbourhood of a specific sample and seeing if they behave the same. In reality, this is analogous to group-wise testing with $n=1$, a very specific type of group testing. In contrast, with SYNG4ME we explore multiple definitions of groups from specific sensitive attributes, to intersectional groups, to points of interest (i.e. $n=1$), to high- and low density regions.
> - AITEST does not account for distributional shift, unlike SYNG4ME which looks at distributional shift with no prior knowledge and high-level knowledge.
>
> **We have included both AITEST and DataSynthesizer in the revised related work to reflect these discussions, see Appendix A.**

---

> ### Author Response · Authors · 2022-11-10
> **Response to reviewer DdMu - Part 3/7**
>
> # (B) Comparison to other tabular data approaches
>
>     "Synthetic Data for Social Good" arXiv:1710.08874 has proposed DataSynthesizer, a privacy-preserving synthetic data generator for creating pathological dataset in tabular domain to be used for model testing. AITEST described in “Data Synthesis for Testing Black-Box Machine Learning Models” (arXiv:2111.02161) also considers group fairness testing as studied in this paper and leverages goal-oriented synthetic data generation in tabular domain for model testing. AITEST has used data constraint-based data synthesis, in addition to using CTGAN and TVAE as baselines, whereas the current work mainly uses CTGAN. Syng4me considers distribution drift for robustness testing whereas AITEST studies adversarial robustness..]
>
> ------
>
> **TL;DR:** We compare to DataSynthesizer and AITEST and showcase differences to SYNG4ME along the following three dimensions: (i) aims and assumptions, (ii) algorithmic, (iii) different use-cases. Updates to Appendix A also reflect this. The side-by-side comparison can be seen in Table 4 (Appendix A) or see https://imgur.com/a/E8FZ5Ou
>
> ------
>
> Thank you for highlighting these two works (DataSynthesizer and AITEST). We contrast SYNG4ME  to both of these works and have **updated Appendix A, to reflect this discussion**. We provide a **detailed side-by-side comparison** of SYNG4ME to work these two works, see https://imgur.com/a/E8FZ5Ou
>
> This new table has also been added to **Appendix A, Table 4**. We provide further details below.
>
>
> ## DataSynthesizer
>
> We believe SYNG4ME is significantly different from DataSynthesizer, in terms of aims, assumptions and algorithmically.
>
> ### Aim and assumptions
> Data Synthesizer primarily focuses on privacy preserving generation of tabular synthetic data.
> The closest component to our work is the extension the paper proposes around adversarial fake data generation. While there are no experiments, the adversarial fake data consists of three areas. We contrast them to SYNG4ME.
>
> The major difference is DataSynthesizer assumes access to **full knowledge about the shift/distributional change**. In contrast, SYNG4ME operates in a different setting - (1) **No prior knowledge** on the shift and (2) **high-level partial knowledge** about the shift through observing some variables in the target domain.
>
> 1. *Edit the distribution*: DataSynthesizer assumes the user has full knowledge of the shift.  SYNG4ME covers two different settings: (1) *No prior knowledge* on the shift, where only minimal assumptions on means of variables allow us to create characteristic curves like in *Section 5.2.1* and (2) *Incorporating prior knowledge*, in which some features are observed from the shifted distribution and we use these to generate the full data from the shifted distribution, like in *Section 5.2.2*. Consequently, the difference is that SYNG4ME tackles the no and partial information settings, whereas DataSynthesizer tackles the full info setting of editing the distribution.
> 2. *Preconfigured pathological distributions* — this requires full and exact knowledge about the shift, which differs from SYNG4ME of partial knowledge and no prior knowledge settings.
> 3. *Injecting missing data, extreme values* — either such an approach is possible to incorporate in SYNG4ME. We see these ideas as complementary.
>
> ### Algorithmic differences
> The authors propose three methods, one with random features, one with independent features and one with correlated features. Due to the absence of correlation in the first two, these reduce the data utility. Let us thus focus on the third method, that does include correlation. This approach uses Bayesian Networks and is only applicable to discrete data, hence needing to discretize continuous variables. This loses utility when a coarse discretization is chosen, while a fine discretization is often intractable and data-inefficient due to the ordinal information being lost, e.g. results for $age=31$ and $age=32$ will generally be similar—exactly the reason why the independent approach was also introduced. Bayesian Nets are also limited in other ways, e.g. results can be influenced by the feature generation order deviating from the real data generation process' ordering, as indicated by the authors of DataSynthesizer in Figures 5 and 6.
>
> ``Continues in next part``

---

> ### Author Response · Authors · 2022-11-10
> **Response to reviewer DdMu - Part 2/7**
>
> ``Continuation: (A) Novelty & contribution of SYNG4ME``
>
> 4. **SYNG4ME formulates distributional shift testing with no shift knowledge to define model sensitivity curves**: Many works assume we have insight into the distributional shift and can edit the distributions of features to assess performance. However, in many cases, we may not know the expected deployment environment or how it may change. In this case, SYNG4ME shows how we can still characterize model behavior and sensitivity for different potential operating conditions. We present model sensitivity curves using synthetic data, which are novel in that they are **analogous to component characteristic curves widely used in engineering fields**. We show that the model sensitivity curves are useful for the following three reasons: (1) *Confident usage*: the synthetic shifts capture the true performance trends as a function of shift, allowing practitioners to understand the models’ expected behavior;  (2) *Insights*: for example, we observe in Section 5.2.1 that shift does not always lead to degradation, challenging the common notion that models always perform poorer when data is shifted;  (3) *Ranking of different available downstream predictive models can help practitioners select models*.
> 5. **SYNG4ME formulates distributional shift testing when we have high-level prior knowledge, as opposed to full knowledge**: We propose a novel formulation of how to incorporate high-level prior knowledge for better distributional shift testing, specifically information or data from the marginal of a subset of the features. This is a realistic scenario in many situations, since high-level statistics of some features (e.g. age and gender) are often freely accessible for a given population. In these instances, we show that the few observed features allow us to accurately generate synthetic data, in turn providing more accurate estimates of a model’s performance on the target domain. As a consequence, we also show that this more accurate estimate of performance allows us to estimate accurately which predictive model will be most accurate for the target domain.
> 6. **SYNG4ME proposes an approach to modeling uncertainty in the performance estimates**: it is essential that we quantify the confidence of SYNG4ME estimates to understand when to trust our performance estimates and when not. SYNG4ME permits quantifying the uncertainty of downstream predictive performance estimates using deep generative ensembles (see **Appendix C.3**).
> 7. **SYNG4ME provides a tool to complete model reports**: Model evaluation reports are important for documenting ML models and their trustworthiness (e.g. see [R11]). **Appendix E** shows how the intersectional performance matrices and distributional shift evaluation, can be used to complete model evaluation reports and characterize model performance. In contrast to prior work, SYNG4ME is low-effort and can be easily customized to a user-provided dataset.

---

> ### Author Response · Authors · 2022-11-10
> **Response to reviewer DdMu - Part 1/7**
>
> Thank you for your thoughtful comments and suggestions. We give answers to each of the following in turn and highlight the updates to the revised manuscript. In addition, we have uploaded the revised manuscript.  We hope this response alleviates your concerns, but please let us know if there are any remaining concerns.
>
> (A) Novelty & contribution of SYNG4ME
>
> (B) Comparison to other tabular data approaches (DataSynthesizer and AITEST)
>
> (C) Novel insights on using synthetic data for testing
>
>
> # (A) Novelty
>
>     The idea of using synthetic data to evaluate ML models is not entirely new.
>
> ------
>
> **TL;DR:** SYNG4ME’s novelty & contribution is across the following dimensions:
> 1. SYNG4ME formulates testing that addresses specific challenges associated with tabular data
> 2. SYNG4ME tackles the specific granular subgroup testing paradigm, where we might have limited real data
> 3. SYNG4ME enables reliable intersectional estimates, proposing the intersectional performance matrix
> 4. SYNG4ME formulates how to generate distributionally shifted test sets without shift knowledge, defining model sensitivity curves
> 5. SYNG4ME formulates how to generate distributionally shifted test sets when we have high-level prior knowledge, as opposed to full knowledge
> 6. SYNG4ME proposes an approach to modeling uncertainty in the performance estimates
> 7. SYNG4ME provides a tool to complete model reports
>
> ------
>
> While testing machine learning models using synthetic data generated via GANs might seem intuitive, we highlight SYNG4ME’s novelty and contributions across the following dimensions. We clarify that the goal of SYNG4ME is different from other approaches: aiming to provide accurate performance estimates (compared to an oracle), compared to real test data alone. Especially, in cases where we may not have lots of real data (subgroups or shifts). In contrast, other approaches primarily focus on probing the weak spots of models, which is less general.
>
> We wish to highlight the potential **impact** of SYNG4ME as a step towards more reliable evaluation of machine learning models—especially in safety-critical, tabular data settings such as healthcare (prostate cancer) and finance (credit scores) which we have examined. We ask the reviewer to consider the importance of providing accurate and reliable estimates of model performance, especially in situations where we might have insufficient real test data. This is especially impactful from a representational bias and fairness perspective— allowing more accurate evaluation of how models will perform, even on minority subgroups or under different distributional settings.
>
> We now delve into the following dimensions of novelty and contributions of SYNG4ME:
>
> 1. **SYNG4ME formulates model testing that addresses specific challenges associated with tabular data**: Most similar work in model testing using synthetic data has involved images. In SYNG4ME, we target tabular data — the most ubiquitous data format in real-world AI (even more so than images and text). *For details, see our response around tabular data*. The focus of SYNG4ME is on tabular data, which means the problem formulation is explicitly different.
> 2. **SYNG4ME tackles the specific granular subgroup testing paradigm, where we might have limited real data**: ML models have been shown to have variable performance for different subgroups [R3-R10]. A challenge, in practice, is how to get reliable performance estimates in cases where we may have access to a limited number of real test examples per subgroup. SYNG4ME addresses this novel use case, by providing more accurate performance estimates to the true oracle (population) estimate of performance, than if we would have used the smaller real test data for model evaluation.
> 3. **SYNG4ME enables reliable intersectional estimates proposing the intersectional performance matrix**: the challenge of reliable performance estimates on small subgroups is even more pronounced on intersectional subgroups, since we may have *insufficient* real test samples to get reliable estimates. We provide novelty in three ways: *(1) Show the value of synthetic data*, in achieving more reliable performance estimates. *(2) Propose the intersectional performance matrix*, an intuitive visual tool to provide model developers insight into where they can improve their model most, as well as inform users how a model may perform on intersections of groups (especially important to evaluate sensitive intersectional subgroups).
>
> ``Continues in next part``

---

> ### Author Response · Authors · 2022-11-15
> **Author follow-up**
>
> Dear Reviewer DdMu,
>
> We are sincerely grateful for your time and energy reviewing the paper.
>
> We hope that our responses and paper updates have addressed your concerns. Please let us know if you have any outstanding concerns—we would be very happy to address these :)
>
> Thank you!
>
> Paper1873 Authors

---

### Official Review · Reviewer_L3gg · 2022-10-27

**Confidence:** 3
**Correctness:** 4
**Technical Novelty And Significance:** 2
**Empirical Novelty And Significance:** 2
**Recommendation:** 5

**Clarity, Quality, Novelty And Reproducibility:**


Clarity and Quality: The paper is clearly written.

Novelty: The problem studied in this paper is interesting, but it has been widely discussed in the machine learning and computer vision community. Meanwhile, using synthetic data for model evaluation also is not a new topic, and the technology used in the paper is not very innovative.

Reproducibility: I think this paper is easy to reproduce.


**Strength And Weaknesses:**

Strength :

1. The problem studied in this paper, i.e., using synthetic data for model evaluation, is interesting and important.

2. An adequate discussion of SYNG4ME use cases is provided, providing some insights into how to use synthetic data for model evaluation.

3. The entire paper is simple to understand. The authors thoroughly review all related work, summarising its benefits and drawbacks.

Weaknesses:

1. Although the problem studied in this paper is intriguing, it is not a novel one that has received much attention in the machine learning and computer vision communities. Using synthetic data to evaluate models is also not a new topic. On the other hand, the technology employed in the paper is not particularly novel.

2. As stated in the paper, SYNG4ME is only capable of handling tabular data. I believe its application scope is too narrow.

**Summary Of The Paper:**

The authors propose in this paper to use synthetic data for model evaluation and to create an automated suite of synthetic data generators (called SYNG4ME). SYNG4ME, in particular, has two main properties: 1) it provides reliable granular evaluation, and 2) it quantifies model sensitivity to distributional shifts. The paper discusses various SYNG4ME cases, which will provide some insights into using synthetic data for model evaluation.

**Summary Of The Review:**

This paper may provide some insights into using synthetic data for model evolution, but I believe it will have little impact on the community (the novelty is limited).

---

> ### Author Response · Authors · 2022-11-10
> **Response to reviewer L3gg - Part 8/8**
>
> ``Continuation of References``
>
> [R22] Suresh, H., & Guttag, J. v. (2019). A Framework for Understanding Sources of Harm throughout the Machine Learning Life Cycle. ACM International Conference Proceeding Series. https://doi.org/10.1145/3465416.3483305
>
> [R23] Cabrera, Á. A., Epperson, W., Hohman, F., Kahng, M., Morgenstern, J., & Chau, D. H. (2019). FairVis: Visual analytics for discovering intersectional bias in machine learning. 2019 IEEE Conference on Visual Analytics Science and Technology (VAST), 46–56.
>
> [R24] Cabrera, Á. A., Epperson, W., Hohman, F., Kahng, M., Morgenstern, J., & Chau, D. H. (2019). FairVis: Visual analytics for discovering intersectional bias in machine learning. 2019 IEEE Conference on Visual Analytics Science and Technology (VAST), 46–56.
>
> [R25] Goel, K., Gu, A., Li, Y., & Ré, C. (2021, August 15). Model Patching: Closing the Subgroup Performance Gap with Data Augmentation. International Conference on Learning Representations. https://doi.org/10.48550/arxiv.2008.06775
>
> [R26] Barocas, S., & Selbst, A. D. (2016). Big data’s disparate impact. Calif. L. Rev., 104, 671.
>
> [R27] Avery, R. B., Brevoort, K. P., & Canner, G. (2012). Does Credit Scoring Produce a Disparate Impact? Real Estate Economics, 40(s1), S65–S114. https://doi.org/https://doi.org/10.1111/j.1540-6229.2012.00348.x
>
> [R28] Bagdasaryan, E., Poursaeed, O., & Shmatikov, V. (2019). Differential Privacy Has Disparate Impact on Model Accuracy. Advances in Neural Information Processing Systems, 32. https://doi.org/10.48550/arxiv.1905.12101
>
> [R29] Timnit Gebru, Jamie Morgenstern, Briana Vecchione, Jennifer Wortman Vaughan, Hanna Wallach, Hal Daume Iii, and Kate Crawford. Datasheets for datasets. ´ Communications of the ACM, 64(12): 86–92, 2021.
>
> [R30] ​​Pang Wei Koh, Shiori Sagawa, Henrik Marklund, Sang Michael Xie, Marvin Zhang, Akshay Balsubramani, Weihua Hu, Michihiro Yasunaga, Richard Lanas Phillips, Irena Gao, et al. Wilds: A benchmark of in-the-wild distribution shifts. In International Conference on Machine Learning. PMLR, 2021.

---

> ### Author Response · Authors · 2022-11-10
> **Response to reviewer L3gg - Part 7/8**
>
> **References**
>
> [R1] Benjelloun, O., Chen, S. and Noy, N., 2020, November. Google dataset search by the numbers. In International Semantic Web Conference (pp. 667-682). Springer, Cham.
>
> [R2] Sun, B., Yang, L., Zhang, W., Lin, M., Dong, P., Young, C. and Dong, J., 2019. Supertml: Two-dimensional word embedding for the precognition on structured tabular data. In Proceedings of the IEEE/CVF Conference on Computer Vision and Pattern Recognition Workshops (pp. 0-0).
>
> [R3] Kaggle Machine Learning & Data Science Survey, 2017
>
> [R4] Shwartz-Ziv, R. and Armon, A., 2022. Tabular data: Deep learning is not all you need. Information Fusion, 81, pp.84-90.
>
> [R5] Cheng, P., Zhu, H., Tang, X., Liu, D., Chen, Y., Wang, X., Pan, W., Ming, Z. and He, X., 2022. DIWIFT: Discovering Instance-wise Influential Features for Tabular Data. arXiv preprint arXiv:2207.02773.
>
> [R6] Borisov, V., Leemann, T., Seßler, K., Haug, J., Pawelczyk, M. and Kasneci, G., 2021. Deep neural networks and tabular data: A survey. arXiv preprint arXiv:2110.01889.
>
> [R7] Qi Wang, Junyu Gao, Wei Lin, and Yuan Yuan. Learning from synthetic data for crowd counting in the wild. In IEEE Conference on Computer Vision and Pattern Recognition (CVPR), pages 8198–8207, 2019.
>
> [R8]  Zheng Tang, Milind Naphade, Stan Birchfield, Jonathan Tremblay, William Hodge, Ratnesh Kumar, Shuo Wang, and Xiaodong Yang. Pamtri: Pose-aware multi-task learning for vehicle re-identification using highly randomized synthetic data. In IEEE International Conference on Computer Vision (ICCV), pages 211–220, 2019
>
> [R9] Daniel Saez Trigueros, Li Meng, and Margaret Hartnett. ´ Generating photo-realistic training data to improve face recognition accuracy. arXiv preprint arXiv:1811.00112, 2018. 2
>
> [R10] Adam Kortylewski, Bernhard Egger, Andreas Schneider, Thomas Gerig, Andreas Morel-Forster, and Thomas Vetter. Analyzing and reducing the damage of dataset bias to face recognition with synthetic data. In Proceedings of the IEEE Conference on Computer Vision and Pattern Recognition Workshops (CVPRW), pages 0–0, 2019
>
> [R11] Qiu, H., Yu, B., Gong, D., Li, Z., Liu, W. and Tao, D., 2021. Synface: Face recognition with synthetic data. In Proceedings of the IEEE/CVF International Conference on Computer Vision (pp. 10880-10890).
>
> [R12] Wei Li, Chengwei Pan, Rong Zhang, Jiaping Ren, Yuexin Ma, Jin Fang, Feilong Yan, Qichuan Geng, Xinyu Huang, Huajun Gong, Weiwei Xu, Guoping Wang, Dinesh Manocha, and Ruigang Yang. Aads: Augmented autonomous driving simulation using data-driven algorithms. CoRR, abs/1901.07849, 2019
>
> [R13] A Gaidon, Q Wang, Y Cabon, and E Vig. Virtual worlds as proxy for multi-object tracking analysis. In CVPR, 2016.
>
> [R14]Weichao Qiu, Fangwei Zhong, Yi Zhang, Zihao Xiao Siyuan Qiao, Tae Soo Kim, Yizhou Wang, and Alan Yuille. Unrealcv: Virtual worlds for computer vision. ACM Multimedia Open Source Software Competition, 2017
>
> [R15] Angel X. Chang, Thomas A. Funkhouser, Leonidas J. Guibas, Pat Hanrahan, Qi-Xing Huang, Zimo Li, Silvio Savarese, Manolis Savva, Shuran Song, Hao Su, Jianxiong Xiao, Li Yi, and Fisher Yu. Shapenet: An information-rich 3d model repository. CoRR, abs/1512.03012, 2015
>
> [R16] Samin Khan, Buu Phan, Rick Salay, and Krzysztof Czarnecki. Procsy: Procedural synthetic dataset generation towards influence factor studies of semantic segmentation networks. In The IEEE Conference on Computer Vision and Pattern Recognition (CVPR) Workshops, June 2019.
>
> [R17] Emanuel Todorov, Tom Erez, and Yuval Tassa. Mujoco: A physics engine for model-based control. 2012 IEEE/RSJ International Conference on Intelligent Robots and Systems, pages 5026–5033, 2012.
>
> [R18] Li, Z. and Xu, C., 2021. Discover the Unknown Biased Attribute of an Image Classifier. In Proceedings of the IEEE/CVF International Conference on Computer Vision (pp. 14970-14979).
>
> [R19] Kortylewski, A., Egger, B., Schneider, A., Gerig, T., Morel-Forster, A. and Vetter, T., 2019. Analyzing and reducing the damage of dataset bias to face recognition with synthetic data. In Proceedings of the IEEE/CVF Conference on Computer Vision and Pattern Recognition Workshops (pp. 0-0).
>
> [R20] McDuff, D., Ma, S., Song, Y. and Kapoor, A., 2019. Characterizing bias in classifiers using generative models. Advances in Neural Information Processing Systems, 32.
>
> [R21] Oakden-Rayner, L., Dunnmon, J., Carneiro, G., & Re, C. (2019). Hidden Stratification Causes Clinically Meaningful Failures in Machine Learning for Medical Imaging. ACM CHIL 2020 - Proceedings of the 2020 ACM Conference on Health, Inference, and Learning, 151–159. https://doi.org/10.48550/arxiv.1909.12475

---

> ### Author Response · Authors · 2022-11-10
> **Response to reviewer L3gg - Part 6/8**
>
>
> ``Continuation - Comparison to testing with synthetic data in computer vision``
>
> **(i) Approaches: CGI- and physics-based simulators**
>
> In Appendix A we have contrasted SYNG4ME to simulators in sequential decision-making and causal inference. Simulators have similarly been used in computer vision to generate synthetic data. The general approach is to prepare the 3D models, place the synthetic model in a controlled scene, set up the environment (camera type, lighting etc.), and render synthetic images to be used for training. This could be done using Unity, Unreal Engine, video games etc. Applications include: self-driving car simulators [R12], object recognition under differing conditions [R13-R15], semantic segmentation [R16], and robotic simulators [R17].
>
> Works on these CGI-type and physics-type simulators **differ** to SYNG4ME, both in aim and goal. These methods are usually used to create (augmented) data for training models—which is different from our goal of testing models. Secondly, an algorithmic difference is that simulators rely on built world models and environments in order to simulate the data, which is in stark contrast to SYNG4ME, which uses generative models to generate data similar to a user-provided dataset.
>
> **(ii) Approaches: Generative models for generating data for probing**
>
> Synthetic data has been used in computer vision for probing models on different dataset attributes. For example, works have generated synthetic data using GANs to probe face recognition algorithms [R18,R19,R20]. For example, do algorithms underperform on specific race groups and can we use synthetic data to probe model blind spots? SYNG4ME’s use cases of synthetic data are different, and we regard existing computer vision synthetic data approaches as complementary rather than overlapping:
> - SYNG4ME aims to generate synthetic data to better approximate true model performance, enabling more accurate model evaluation for subgroups or intersectional subgroups for which we have **limited real test data**.  This is different from computer vision applications, in which the aim is also to find model weaknesses, but limited data to train the generative model is not an issue.
> - SYNG4ME aims to generate synthetic data that tests models under shifts in distribution. When no prior knowledge is present, we aim to produce performance characteristic curves across the model's operating range. Of course, this could highlight blind spots, however the goal is to produce an operating curve and understand model behavior across the board. Secondly, SYNG4ME assesses the case where we have no full data from the target, only limited information about marginals. This idea of testing models under distributional shifts and incorporating knowledge is different from computer vision applications, which might instead (i) aim to find other datasets in the wild [R30], or (ii) use tools such as style transfer to test distributional shifts. Finding related datasets is often not possible for tabular data, due to feature mismatch. Style transfer does not necessarily extend to tabular data, due to content and style being not clearly disentangled in tabular data, there often not being enough data to train such a model, and style transfer does not allow incorporating prior shift knowledge.
>
>
> *We hope this response alleviates your concerns. We have uploaded the revised manuscript, which includes an extended related works section. Please let us know if there are any remaining concerns.*

---

> ### Author Response · Authors · 2022-11-10
> **Response to reviewer L3gg - Part 5/8**
>
> # (C) Comparison to testing with synthetic data in computer vision
>
>     [Synthetic data] received much attention in the machine learning and computer vision communities]
>
> ------
> **TL;DR:**  We contrast works in Computer Vision - both simulators and generative models to SYNG4ME. Updates to Appendix A also reflect this. The detailed side-by-side comparison is shown in Table 3 (Appendix A) - or see https://imgur.com/a/bnFsXZO
>
> ------
>
> Thank you for mentioning the use of synthetic in areas such as computer vision. We have discussed the differences between simulators in sequential-decision making and causal inference in **Appendix A**. We have **updated** the extended related work in **Appendix A** to reflect the discussion below on computer vision applications.
>
> As mentioned by the reviewer, synthetic data has been explored in the computer vision community. Examples include crowd counting [R7], vehicle identification for self-driving cars [R8], and face recognition [R9,R10,R11].
>
> We provide a **detailed side-by-side comparison** of SYNG4ME to work in computer vision, see https://imgur.com/a/bnFsXZO
>
> This new table has also been added to **Appendix A, Table 3**.
>
> We highlight additional points below.
>
> These methods can be grouped as follows by their motivations:
> - Generate synthetic data for training, to reduce the reliance on collecting and annotating large training sets—*This is different from SYNG4ME as they focus on constructing better training sets, rather than constructing better test sets for model evaluation*
> - Generate synthetic data to improve the model by augmenting the real dataset with synthetic examples—*Again, this is different from SYNG4ME as they focus on training better-performing models, rather than the evaluation of an already trained model*
> - Generate synthetic data to probe models on different dataset attributes. For example, in face recognition, how the model might perform on faces with long vs short hair. This is most similar to SYNG4ME, but there are clear differences in both the goal for and approach to generating the data. We compare SYNG4ME to (i)  CGI- or physics-based simulators and (ii) deep generative models for probing in computer vision.
>
> ``Continues in next part``

---

> ### Author Response · Authors · 2022-11-10
> **Response to reviewer L3gg - Part 4/8**
>
> ``Continuation: (B) Novelty & contribution of SYNG4ME``
>
> 4. **SYNG4ME formulates distributional shift testing with no shift knowledge to define model sensitivity curves**: Many works assume we have insight into the distributional shift and can edit the distributions of features to assess performance. However, in many cases, we may not know the expected deployment environment or how it may change. In this case, SYNG4ME shows how we can still characterize model behavior and sensitivity for different potential operating conditions. We present model sensitivity curves using synthetic data, which are novel in that they are **analogous to component characteristic curves widely used in engineering fields**. We show that the model sensitivity curves are useful for the following three reasons: (1) *Confident usage*: the synthetic shifts capture the true performance trends as a function of shift, allowing practitioners to understand the models’ expected behavior;  (2) *Insights*: for example, we observe in **Section 5.2.1** that shift does not always lead to degradation, challenging the common notion that models always perform poorer when data is shifted;  (3) *Ranking of different available downstream predictive models can help practitioners select models*.
> 5. **SYNG4ME formulates distributional shift testing when we have high-level prior knowledge, as opposed to full knowledge**: We propose a novel formulation of how to incorporate high-level prior knowledge for better distributional shift testing, specifically information or data from the marginal of a subset of the features. This is a realistic scenario in many situations, since high-level statistics of some features (e.g. age and gender) are often freely accessible for a given population. In these instances, we show that the few observed features allow us to accurately generate synthetic data, in turn providing more accurate estimates of a model’s performance on the target domain. As a consequence, we also show that this more accurate estimate of performance allows us to estimate accurately which predictive model will be most accurate for the target domain.
> 6. **SYNG4ME proposes an approach to modeling uncertainty in the performance estimates**: it is essential that we quantify the confidence of SYNG4ME estimates to understand when to trust our performance estimates and when not. SYNG4ME permits quantifying the uncertainty of downstream predictive performance estimates using deep generative ensembles (see **Appendix C.3**).
> 7. **SYNG4ME provides a tool to complete model reports**: Model evaluation reports are important for documenting ML models and their trustworthiness (e.g. see [R29]). **Appendix E** shows how the intersectional performance matrices and distributional shift evaluation, can be used to complete model evaluation reports and characterize model performance. In contrast to prior work, SYNG4ME is low-effort and can be easily customized to a user-provided dataset.

---

> ### Author Response · Authors · 2022-11-10
> **Response to reviewer L3gg - Part 3/8**
>
> # (B) Novelty & contribution of SYNG4ME
>
> ------
> **TL;DR:** SYNG4ME’s novelty & contribution is across the following dimensions:
> 1. SYNG4ME formulates testing that addresses specific challenges associated with tabular data
> 2. SYNG4ME tackles the specific granular subgroup testing paradigm, where we might have limited real data
> 3. SYNG4ME enables reliable intersectional estimates, proposing the intersectional performance matrix
> 4. SYNG4ME formulates how to generate distributionally shifted test sets without shift knowledge, defining model sensitivity curves
> 5. SYNG4ME formulates how to generate distributionally shifted test sets when we have high-level prior knowledge, as opposed to full knowledge
> 6. SYNG4ME proposes an approach to modeling uncertainty in the performance estimates
> 7. SYNG4ME provides a tool to complete model reports
> ------
>
> While testing machine learning models using synthetic data generated via GANs might seem intuitive, we highlight SYNG4ME’s novelty and contributions across the following dimensions. We clarify that the goal of SYNG4ME is different from other approaches: aiming to provide accurate performance estimates (compared to an oracle), compared to real test data alone. Especially, in cases where we may not have lots of real data (subgroups or shifts). In contrast, other approaches primarily focus on probing the weak spots of models, which is less general.
>
> We wish to highlight the potential **impact** of SYNG4ME as a step towards more reliable evaluation of machine learning models—especially in safety-critical, tabular data settings such as healthcare (prostate cancer) and finance (credit) which we have examined. We ask the reviewer to consider the importance of providing accurate and reliable estimates of model performance, especially in situations where we might have insufficient real test data. This is especially impactful from a representational bias and fairness perspective— allowing more accurate evaluation of how models will perform, even on minority subgroups or under different distributional settings.
>
> We now delve into the following dimensions of novelty and contributions of SYNG4ME:
>
> 1. **SYNG4ME formulates model testing that addresses specific challenges associated with tabular data**: Most similar work in model testing using synthetic data has involved images. In SYNG4ME, we target tabular data — the most ubiquitous data format in real-world AI (even more so than images and text). *For details, see our response around tabular data*. The focus of SYNG4ME is on tabular data, which means the problem formulation is explicitly different.
> 2. **SYNG4ME tackles the specific granular subgroup testing paradigm, where we might have limited real data**: ML models have been shown to have variable performance for different subgroups [R21-R28]. A challenge, in practice, is how to get reliable performance estimates in cases where we may have access to a limited number of real test examples per subgroup. SYNG4ME addresses this novel use case, by providing more accurate performance estimates to the true oracle (population) estimate of performance, than if we would have used the smaller real test data for model evaluation.
> 3. **SYNG4ME enables reliable intersectional estimates proposing the intersectional performance matrix**: the challenge of reliable performance estimates on small subgroups is even more pronounced on intersectional subgroups, since we may have *insufficient* real test samples to get reliable estimates. We provide novelty in three ways: *(1) Show the value of synthetic data*, in achieving more reliable performance estimates. *(2) Propose the intersectional performance matrix*, an intuitive visual tool to provide model developers insight into where they can improve their model most, as well as inform users how a model may perform on intersections of groups (especially important to evaluate sensitive intersectional subgroups).
>
> ``Continues in next part``

---

> ### Author Response · Authors · 2022-11-10
> **Response to reviewer L3gg - Part 2/8**
>
> # (A) Applications of tabular data are narrow
>
>     SYNG4ME is only capable of handling tabular data. I believe its application scope is too narrow.]
>
> ------
> **TL;DR:** Tabular data is the most common and ubiquitous data format across many real-world applications compared to other modalities. It is imperative that we build tools to test ML systems with real-world impact reliably. SYNG4ME provides such a solution for tabular data that is applicable to the majority of real-world ML problems.
>
> ------
>
> While the focus on tabular data is seemingly limiting, the reality is that **tabular data accounts for the majority of real-world ML applications**—for which reliable testing of models is critical. More specifically, in real-world applications, tabular data is the most common and ubiquitous data format across many fields, such as medicine, finance, manufacturing, e-commerce and climate science, where data is based on relational databases [R2,R4,R5,R6]. This is further highlighted as the Google Dataset platform has around **67% of its 12 million datasets as structured/tabular data** [R1] - which our paper addresses. In addition, Kaggle’s survey of over 16000 data professionals (data scientists, software engineers, data analysts) [R2,R3] found that at least **65% work with tabular data on a daily basis**. In comparison, other modalities are significantly lower, such as images (18%), video (5%), other (11%) etc. If we just consider data scientists, this difference is even more prominent, with **79% working on tabular data daily vs. around 14% on images**.
>
> Based on the aforementioned importance and ubiquity of tabular data in real-world ML, it is crucial to ensure that real-world systems that operate with tabular data are reliably tested. SYNG4ME provides a novel contribution for doing so, hence we believe it is relevant for the majority of real-world ML problems.

---

> ### Author Response · Authors · 2022-11-10
> **Response to reviewer L3gg - Part 1/8**
>
> Thank you for your thoughtful comments and suggestions. We give answers to each of the following in turn and highlight the updates to the revised manuscript. In addition, we have uploaded the revised manuscript. We hope this response alleviates your concerns, but please let us know if there are any remaining concerns.
>
> **(A) Applications of tabular data are narrow**
>
> **(B) Novelty**
>
> **(C) Comparison to testing with synthetic data in computer vision**

---

> ### Author Response · Authors · 2022-11-15
> **Author follow-up**
>
> Dear Reviewer L3gg,
>
> We are sincerely grateful for your time and energy reviewing the paper.
>
> We hope that our responses and paper updates have addressed your concerns. Please let us know if you have any outstanding concerns—we would be very happy to address these :)
>
> Thank you!
>
> Paper1873 Authors

---

### Author Response · Authors · 2022-11-10
**Response Overview**

   We thank the reviewers for their insightful and positive feedback!

We are encouraged that they found SYNG4ME’s usage of synthetic data to evaluate machine learning models more carefully and w.r.t. distributional shifts is both “timely and critical” (**R-DdMu**), addressing an “interesting and important” (**R-L3gg, R-NW99**) problem, which is “novel and beneficial to the research community to ensure reliability and robustness of evaluation” (**R-kuUj**).

We are glad that they found the proposed method “technically sound”(**R-NW99**), with the experiments and analysis of the use-cases of granular evaluation and distributional shifts “providing some insights into how to use synthetic data for model evaluation”(**R-L3gg**) , especially w.r.t. the issues of evaluation on small test sets being deemed “convincing”(**R-kuUj**)

We address specific questions and concerns below and have incorporated all feedback by highlighting updates to the revised manuscript and supplementary material, which we have uploaded.

**On the basis of our clarifications and updates, we hope we have addressed the reviewers' concerns. If any concerns remain, please let us know.**

Thank you for your kind consideration!

---

> ### Author Response · Authors · 2022-11-15
> **Summary of changes**
>
> Thank you again to the reviewers for their generous comments and suggestions. We hope we have satisfied all remaining concerns.
>
> In addition to our responses (10 November), we have also updated the paper based on the reviewers' comments, which we believe has improved the paper.
>
> In addition, we provide new experimental results below (15 November), which we hope will be helpful.
>
> The changes are summarized as follows:
>
> **New experiments**
>
> * We have conducted an additional distributional shift experiment to illustrate how SYNG4ME could be used to understand the effect of distribution shift by incorporating prior knowledge. We conduct this experiment on Covid-19 data from Brazil. We wish to understand potential model performance trained in the north of the country if we then deploy the model to the south of the country - where there is a known distribution shift as a result of the different populations across regions. For example: different prevalence of respiratory issues, sex proportions, obesity rates etc. We have included a **new Appendix D.5** to reflect this. For a snapshot see https://imgur.com/a/YFZd5Li
>
> * We have conducted an additional subgroup experiment to illustrate how SYNG4ME can provide more reliable estimates of fairness metrics (disparate impact and equalized odds) when compared to using smaller test data alone. We have included a **new Appendix D.4** to reflect this. This result is impactful from a representational bias and fairness perspective.
>
> **Clarifications**
>
> * Clarified and contrasted the difference of SYNG4ME’s usage of synthetic data to computer vision; **see Appendix A and Table 4**
> * Clarified and contrasted the difference of SYNG4ME’s usage of synthetic data to DataSynthesizer and AITEST; **see Appendix A and Table 5**
>
> *With our responses and paper updates, we hope that we have addressed the reviewers' concerns. Please let us know if there were any further questions or comments. We are eager to do our utmost to address them!*
>
> Thank you for your kind consideration :)
>
> Paper1873 Authors

---

### Author Response · Authors · 2022-12-05
**SYNG4ME's relevance as highlighted at NeurIPS 2022**

Dear Reviewers and Area Chair,

Thank you again for your time and invaluable feedback during the reviewing process!

We wish to highlight the importance of the problem tackled by SYNG4ME: model evaluation and testing, which was emphasized as a key open challenge for the community at NeurIPS 2022.

We discuss below how SYNG4ME addresses the challenges outlined by the following talks at NeurIPS:
- Isabelle Guyon, NeurIPS 2022 keynote (e.g. slides 41 & 98) [[Link]](https://neurips.cc/virtual/2022/invited-talk/56158), also accessible [here](https://sites.google.com/chalearn.org/mldb/home?pli=1) (e.g. slides 16 & 29): highlighted the importance of new approaches to stress testing, in terms of trust, robustness, and blind spots of ML models, and how this is a critical component of reproducible science.
    - SYNG4ME helps address these challenges as follows. (1) more reliable evaluation of subgroups (in terms of accuracy and fairness) (Section 5.1 \& Appendix D.4). (2) the intersectional performance matrix (Figure 4) and distribution shift (Section 5.2) experiments provide a more in-depth understanding of where the model performs well and where it fails (i.e. addressing the challenge of robustness and blind spots of ML models).
- J.P. Morgan, SyntheticData4ML workshop (Slide 2) [[Link]](https://neurips.cc/virtual/2022/workshop/50016): emphasized the need in sensitive applications, such as finance, to test that ML models reliably handle a wide variety of potential input situations.
    - SYNG4ME allows users to assess how ML models respond to different inputs, for example shifts across their operating range (Section 5.2).

We appeal to the reviewers to consider the timeliness and importance of the problem tackled by SYNG4ME.

We hope that our responses and paper updates have addressed any concerns and we would sincerely appreciate it if you would consider updating your score and assessment of our paper.

Thank you!

Paper1873 Authors

---

### Decision · Program_Chairs · 2023-01-20

**Decision:**

Reject

**Justification For Why Not Higher Score:**

The limited novelty of paper, the focus only on tabular data, and the empirical evaluation on small datasets preclude me from justifying a higher score.

**Justification For Why Not Lower Score:**

N/A

**Metareview: Summary, Strengths And Weaknesses:**

The authors propose SYNG4ME: a method of evaluating models by leveraging synthetic datasets. This idea is not new and there exists much prior work on this type of evaluation. What is novel is the focus on tabular data, which is comparatively less studied compared to domains such as images, text, and speech. The reviewers, however, found this limited scope more a weakness of the paper rather than a strength. Furthermore, the authors propose the empirical evaluation only on relatively simple datasets. Finally, the limited technical novelty was something mentioned in a number of reviews.

Extensions to other modalities, and evaluation on larger datasets would greatly improve the paper.